# Differential regulation of alternative promoters emerges from unified kinetics of enhancer-promoter interaction

Jingyao Wang [1,2], Shihe Zhang [2,3✉], Hongfang Lu[1,2] & Heng Xu [2,3✉]

Many eukaryotic genes contain alternative promoters with distinct expression patterns. How these promoters are differentially regulated remains elusive. Here, we apply single-molecule imaging to quantify the transcriptional regulation of two alternative promoters (P1 and P2) of the Bicoid (Bcd) target gene *hunchback* in syncytial blastoderm *Drosophila* embryos. Contrary to the previous notion that Bcd only activates P2, we find that Bcd activates both promoters via the same two enhancers. P1 activation is less frequent and requires binding of more Bcd molecules than P2 activation. Using a theoretical model to relate promoter activity to enhancer states, we show that the two promoters follow common transcription kinetics driven by sequential Bcd binding at the two enhancers. Bcd binding at either enhancer primarily activates P2, while P1 activation relies more on Bcd binding at both enhancers. These results provide a quantitative framework for understanding the kinetic mechanisms of complex eukaryotic gene regulation.

[1] School of Life Sciences and Biotechnology, Shanghai Jiao Tong University, 200240 Shanghai, China. [2] Institute of Natural Sciences, Shanghai Jiao Tong University, 200240 Shanghai, China. [3] School of Physics and Astronomy, Shanghai Jiao Tong University, 200240 Shanghai, China. ✉email: hsx_sjtu@sjtu.edu.cn; Heng_Xu@sjtu.edu.cn

Promoters initiate gene transcription by interacting with specific *cis*-regulatory sequences (enhancers)[1–3]. In eukaryotic genomes, many genes contain alternative promoters[4,5], which can produce functionally distinct transcript isoforms under the regulation of multiple enhancers[6,7]. During development, the differential expression of these isoforms is critical to cell fate decisions[8,9]. Misregulation of these isoforms can lead to diseases, including cancer[6,10–12]. Thus, eukaryotic gene regulation needs to be understood at the level of individual alternative promoters and enhancers.

Enhancer activation of a single promoter involves a series of molecular events, including transcription factor (TF) binding, local chromatin opening, and physical proximity between the two elements[1,2,13]. Typically, TF binding determines the spatial expression pattern[1,14], whereas other events set the expression amplitude[15–18]. When multiple promoters and enhancers are present, interactions between these elements often result in complex and varied promoter behaviors[19]. For example, multiple enhancers may contact each other and synergistically drive a promoter[20,21]. Alternatively, different enhancers may be mutually exclusive and compete for promoter activation[22,23]. In these cases, enhancers combine their regulatory effects differently, ranging from superadditive to subadditive or even repressive summation[12]. Similarly, a single enhancer may activate multiple promoters simultaneously[24,25] or one at a time[26]. So far, although specific mechanisms were proposed for some of these phenomena, it is unclear whether universal mechanisms exist for the complex interactions between multiple enhancers and promoters.

An ideal model to investigate alternative promoter regulation is the *Drosophila* gap gene *hunchback* (*hb*), which contains two alternative promoters (P1 and P2, Fig. 1a)[27–30]. In early embryogenesis (syncytial blastoderm stage, nuclear division cycles (nc) 10–13), *hb* is expressed in a bursty manner[31,32] throughout the anterior half of the embryo in response to the concentration gradient of the maternal TF Bicoid (Bcd)[29,33,34]. This expression is believed to be purely from the P2 promoter mediated by two enhancers: a proximal enhancer located next to P2[29,34] and a distal shadow enhancer located 4.5 kb upstream[35]. Cooperative Bcd binding at either enhancer can activate P2[35]. Competitive action of these partially redundant enhancers helps suppress expression noise and ensure a robust expression pattern[23,35].

Unlike P2, the P1 promoter is believed to be inactive during early development[36,37], due to its lack of Zelda binding sites and TATA box necessary for local chromatin opening[30,38]. Instead, the promoter is activated in late nc14 by a Bcd-independent stripe enhancer[28,36]. However, previous measurements of endogenous P1 activity relied on traditional in situ hybridization methods, which have limited sensitivity to detect weak mRNA signals. In fact, *hb*-reporter experiments showed that P1 could respond to Bcd when placed adjacent to the proximal or distal enhancers[30]. Without precise quantification of endogenous P1 activity, the understanding of early *hb* regulation is incomplete. It is unclear how P1 interacts with different *hb* enhancers and whether the mechanisms of P1 and P2 regulation are intrinsically related.

Here, we use single-molecule fluorescence in situ hybridization (smFISH)[31,32,39,40] to quantify the expression of individual P1 and P2 promoters for each endogenous *hb* gene locus in nc11–13 embryos. Contrary to the previous notion, we find that P1 contributes a modest yet non-negligible fraction of early *hb* transcription and affects the expression patterns of other gap genes. Using different TF dosages and enhancer deletion, we show that Bcd activates both promoters via the proximal and distal enhancers. Compared with P2, P1 activation requires cooperative binding of more Bcd molecules and a synergistic (as opposed to competitive) action of the two enhancers. Analyzing the statistics of nascent mRNA signals from individual promoter loci reveals

that both promoters follow a unified scheme of three-state transcription kinetics. Cooperative Bcd binding at either enhancer can drive a promoter to a weak active state, while additional Bcd binding at the second enhancer can turn the promoter to full-power transcription. The two promoters differ in their responses to different Bcd binding configurations. P2 transcription is primarily driven by Bcd binding at a single enhancer, while P1 transcription relies more on Bcd binding at both enhancers. In concert, these results provide a simple and quantitative mechanism for the differential regulation of alternative promoters. Our quantitative approach may be generalized as a framework for deciphering complex eukaryotic gene regulation involving multiple promoters and enhancers.

## Results

### *hb* P1 and P2 promoters are both active in early embryogenesis.

Each *hb* promoter produces a unique transcript isoform (P1: *hb*-RB, P2: *hb*-RA; Supplementary Fig. 1a–c; see Supplementary Note 1)[29,41]. To quantify the endogenous P1 and P2 activities in early embryos, we applied smFISH with four sets of oligonucleotide probes designed for different regions of *hb* mRNAs. Specifically, two P1-specific probe sets targeted the 5′ untranslated region (UTR) and intron region of *hb*-RB. A P2-specific probe set targeted the 3′UTR of *hb*-RA. Finally, a probe set targeted the coding sequence (CDS) region shared by *hb*-RA and *hb*-RB (Fig. 1a and Supplementary Table 1).

Confocal imaging and automated image analysis identified actively transcribing *hb* loci as bright FISH spots in wild-type (WT) syncytial blastoderm embryos (Fig. 1b). We quantified the instantaneous transcriptional activity of every locus in units of individual cytoplasmic mRNAs[32] (Fig. 1b and Supplementary Fig. 2; see Methods section). For the CDS signal, we observed bright FISH spots in the anterior part of nc11–13 embryos (Fig. 1c). Most anterior nuclei contained two bright spots, while some nuclei exhibited three or four spots due to the replication of the *hb* gene (Supplementary Fig. 3a). These results are consistent with literature[31], indicating that at least one of the two *hb* promoters is active in the anterior side of the embryo.

Similar to the CDS signal, promoter-specific FISH signals were also concentrated in the anterior half of nc11–13 embryos (Fig. 1c). Specifically, ~70% of the anterior nuclei (within the range of 0.20–0.40 embryo length (EL)) showed bright P2–3′UTR spots (Fig. 1d), consistent with previous reports of active P2 expression in early development[27,30]. Surprisingly, ~32%–57% of the anterior nuclei in nc11–13 embryos also contained bright P1–5′UTR and intron spots (Fig. 1d), indicating early P1 transcription, a phenomenon barely observed in previous bulk studies (Supplementary Fig. 1d, e). The percentage of P1-active anterior nuclei increased with the nuclear cycle (Fig. 1d), suggesting that P1 becomes increasingly active during development. In total, P1- and P2-specific probe signals were exhibited in ~73%–84% of the anterior nuclei, lower than that of the CDS signal. This percentage difference may be because the P2-specific probes target the very end of *hb*-RA, which is missing in many incomplete nascent transcripts.

To analyze the instantaneous transcription of P1 and P2, we plotted, for each probe set, the nascent mRNA signal against the nuclear position for all nuclei in the embryo (Fig. 1e and Supplementary Fig. 3b). During nc11–13, the average expression profile for each probe set exhibited a reverse-sigmoidal shape within the range 0.20–0.70 EL. By fitting the profile to a logistic function, we estimated the maximal signal level $r_{max}$ and the boundary position $x_0$ of the anterior expression domain (Supplementary Fig. 3b; see Methods section). As expected, $r_{max}$ of P1-specific signals was much smaller than that of the CDS

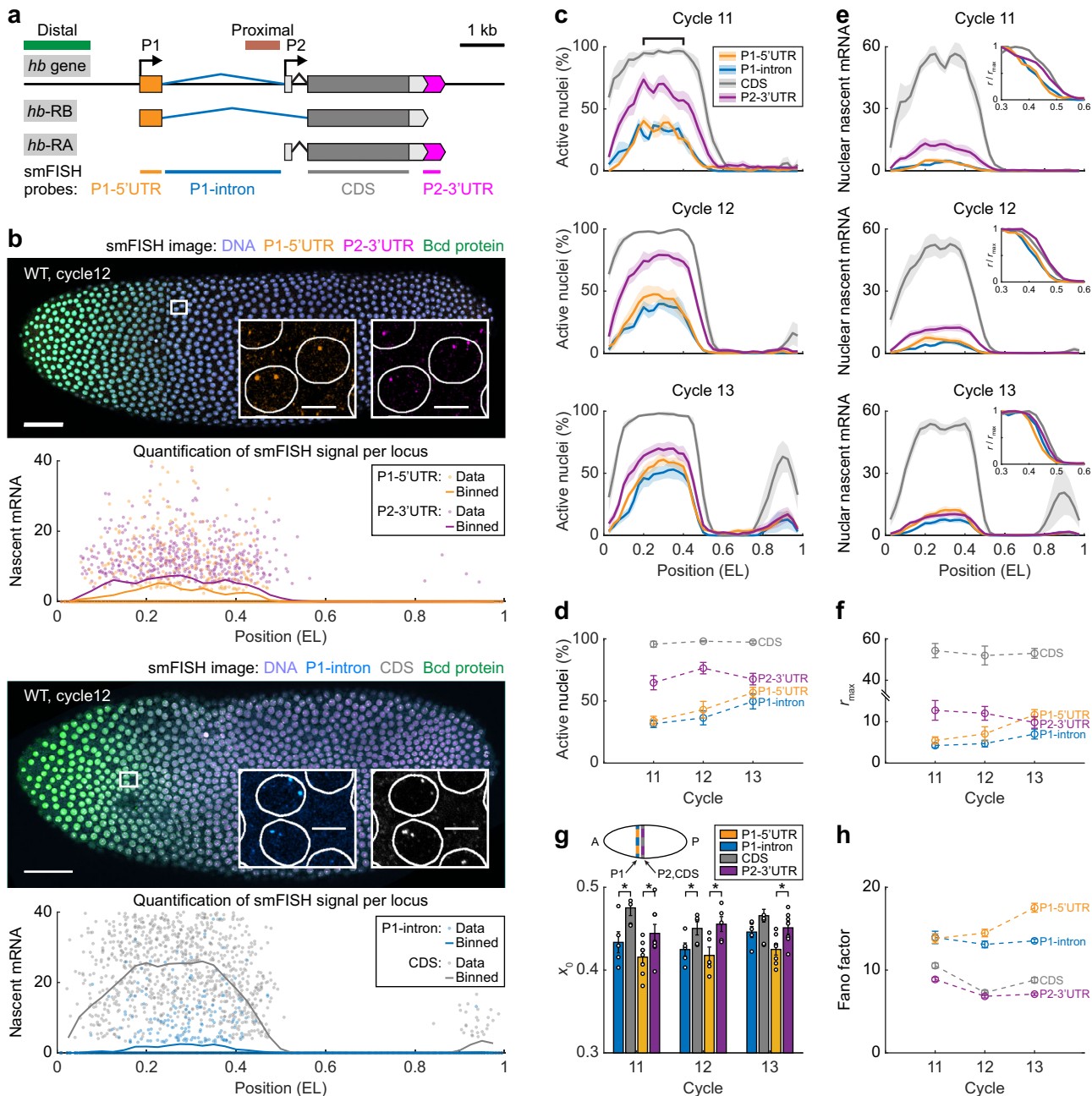

**Fig. 1 Absolute quantification of *hb* transcription reveals P1 and P2 activities in early embryogenesis. a** Schematic of the endogenous *hb* locus with two Bcd-dependent enhancers (green: distal enhancer, brown: proximal enhancer) and two promoters (P1 and P2). Four smFISH probe sets were used to label different regions of the two *hb* mRNA isoforms. **b** Confocal images of two WT *Drosophila* embryos, each labeled for two regions of *hb* mRNAs, Bcd protein, and DNA at nc12. Scale bars, 50 μm. The experiment was repeated three times, independently, with similar results. Insets, magnified views of anterior nuclei. Scale bars, 5 μm. Nascent signal at individual *hb* loci (in units of the number of mRNA molecules) was plotted against the anterior-posterior (AP) position for different probe sets. The single-locus data were binned along the AP axis (bin size: 0.05 EL, step size: 0.025 EL). **c** Percentage of active nuclei as a function of the AP position for different probe signals during nc11–13. Marked region, 0.2–0.4 EL. Shadings indicate s.e.m. **d** Average percentage of active nuclei in the position range of 0.2–0.4 EL for different probe signals during nc11–13. **e** Nascent signal per nucleus (in units of the number of mRNA molecules) as a function of the AP position for different probe sets during nc11–13. Shadings indicate s.e.m. Insets, expression profiles were scaled by the maximal signal levels. **f**, **g** The maximal signal level (**f**) and the boundary position (**g**) of the anterior expression domain for different probe sets during nc11–13, with two-sided *t*-test in **g** (P1-Intron vs. CDS: $p = 0.03$, 0.047, and 0.087 for nc11–13, respectively; P1–5'UTR vs. P2–3'UTR: $p = 0.045$, 0.024, and 0.024 for nc11–13, respectively. *$p < 0.05$). **h** The Fano factor for the nascent signal at individual *hb* gene loci in the position range of 0.2–0.4 EL for different probe sets during nc11–13. **c**–**h** Data are presented as mean ± s.e.m. P1–5'UTR and P2–3'UTR: $n = 8$, 5, and 7 biologically independent embryos at nc11–13, respectively; P1-Intron and CDS: $n = 5$ biologically independent embryos at each nuclear cycle. The spatial profile of each embryo was binned from the single-nucleus data (bin size: 0.05 EL, step size: 0.025 EL). Source data are provided as a Source Data file.

signal (Fig. 1f), indicating that P1 contributes less than P2 to early *hb* expression. $r_{max}$ of the P2-specific signal also remained low (Fig. 1f) due to the probes' target location at the 3′ end of *hb*-RA. $r_{max}$ of the P1- and P2-specific signals should not be directly compared, since their probes target different ends of nascent transcripts (5′ vs. 3′; see next section for a comparison between P1- and P2-specific signal levels). The boundary position $x_0$ of the P1-specific signals was significantly lower than that of the P2-specific and CDS signals (~0.42 EL v.s. ~0.46 EL, Fig. 1e, g), indicating that P1 is activated at a more anterior position than P2.

In addition to the mean expression level, the activities of individual promoter loci exhibited substantial variability (Fig. 1b). The Fano factors of anterior P1- and P2-specific nascent signals were much larger than one (Fig. 1h; see Methods section), indicating bursty transcription from both promoters[42,43]. Such burstiness primarily resulted from the intrinsic stochasticity of *hb* transcription, as the two promoters behaved independently (Supplementary Fig. 3c, d; see Methods section). Quantifying the intrinsic noise[44] revealed that P1 was much noisier than P2 (Supplementary Fig. 3e; see Methods section), consistent with its low expression level.

**P1 contributes a non-negligible fraction of early *hb* transcription and function**. To quantify the contributions of P1 and P2 to early *hb* transcription, we note that the *hb* CDS signal reflects either P1 or P2 transcription. Using CRISPR-based mutant fly lines with P1 and P2 deletions (ΔP1C and ΔP2C)[30], we showed that the average nascent CDS signal from either promoter was proportional to the signal of promoter-specific probes (Supplementary Fig. 4 and see Supplementary Note 3). Thus, the nascent CDS signal ($r_{CDS}$) in the WT embryo is a linear combination of the nascent P1–5′UTR and P2–3′UTR signals ($r_{P1}$ and $r_{P2}$):

$$r_{CDS} = a_1 r_{P1} + a_2 r_{P2} \qquad (1)$$

where $a_1$ and $a_2$ are ratio parameters that, in theory, depend on the mRNA elongation and termination dynamics (see Supplementary Note 6)[45]. The two terms on the right-hand side of the equation distinguish the contribution of each promoter to the CDS signal.

To examine Eq. (1), we compared nuclear P1–5′UTR, P2–3′UTR, and CDS signals in the same embryo (Fig. 2a; see Methods). The three signals satisfied a linear relationship (Fig. 2b) with $a_1 = 0.53 \pm 0.03$ and $a_2 = 3.72 \pm 0.90$ (Fig. 2c). In a simple transcription model with deterministic elongation and termination processes[45,46], the above $a_1$ and $a_2$ values indicate a post-elongation residence time ($T_R$) in terminating a nascent mRNA (Fig. 2d; Supplementary Fig. 5b; and see Supplementary Note 6). The estimated $T_R$ for P1 and P2 was 142 s and 22 s, respectively, consistent with previous estimations of transcription termination[45,47,48]. These results validate Eq. (1) and enable the decomposition of the nascent CDS signal into different promoter activities (Supplementary Fig. 5c; see Methods section). In the anterior expression domain, the contribution of P1 increased with the nuclear cycle to ~13% (Fig. 2e), which was modest yet non-negligible.

In addition to the mean expression level, we investigated the contributions of P1 and P2 to the intrinsic noise of nascent *hb* transcription. Considering that P1 and P2 activities were independent, their contributions to CDS noise ($\eta^2_{CDS}$) could be summed as:

$$\eta^2_{CDS} = \eta^2_{P1-CDS} f^2_{P1-CDS} + \eta^2_{P2-CDS} f^2_{P2-CDS} \qquad (2)$$

where $\eta^2_{P1-CDS}$ and $\eta^2_{P2-CDS}$ denote CDS noises originating from P1 and P2 activities, respectively, while $f_{P1-CDS}$ and $f_{P2-CDS}$

indicate the relative contributions of each promoter to the mean CDS signal (see Supplementary Note 6). Equation (2) revealed that P1 contributed to up to ~20% of the anterior CDS noise during nc11–13 (Fig. 2f), which exceeded its contribution to the mean CDS signal.

To evaluate the function of early P1 transcription in embryo development, we measured the expression patterns of *hb*-target genes, *Krüppel* (*Kr*) and *knirps* (*kni*), in P1 deletion embryos (ΔP1C)[30]. Based on the literature, the anterior boundaries of *Kr* and *kni* expression domains in nc14 are both set by the repression of Hb protein[49,50]. Consistent with this picture, the anterior *Kr* and *kni* boundaries shifted anteriorly for ~0.09 EL (from $0.45 \pm 0.02$ EL in WT to $0.36 \pm 0.01$ EL in ΔP1C) and ~0.06 EL (from $0.58 \pm 0.01$ EL in WT to $0.52 \pm 0.02$ EL in ΔP1C), respectively, in response to P1 deletion (Fig. 2g, h). Thus, early P1 transcription has a non-negligible effect on *Drosophila* AP patterning. However, no obvious cuticle disruption was reported for ΔP1C larvae[30], suggesting compensation from other genes in the downstream patterning process.

**Activating P1 requires cooperative binding of more Bcd molecules than activating P2**. In early embryogenesis, the exponential gradient of Bcd is the primary driver of anterior *hb* transcription[29,34]. To examine whether Bcd directly regulates both promoters, we used a transgenic fly line with 1× functional *bcd* gene[51]. With reduced Bcd dosage, the anterior expression domains of P1 and P2 both retreated towards the anterior pole (Fig. 3a). Specifically, P1 and P2 expression boundaries in 1× *bcd* embryos significantly shifted to the anterior side by ~0.07 EL (Fig. 3b). Thus, Bcd activates both promoters in nc11–13 embryos.

To quantify the regulation of each promoter by Bcd, we combined promoter-specific smFISH of *hb* mRNA with immunofluorescence of Bcd (Fig. 1b). We estimated the absolute Bcd concentration in each nucleus using a previously developed image analysis method[32] (Supplementary Fig. 6a, b; see Methods section and Supplementary Note 4). For each promoter, we plotted the average transcriptional response per nucleus (0.25–0.75 EL) to Bcd concentration in the embryo (Fig. 3c), known as the gene regulation function (GRF)[52]. Previous studies reported that *hb* GRF fitted well to a Hill function with a Hill coefficient $h \approx 5–6$[32,53]. A common explanation was that Bcd activates *hb* by cooperatively binding multiple sites in the regulatory sequence[33,54,55]. Here, we found that the Hill coefficient of P1-specific GRFs (P1–5′UTR: $h = 7.1 \pm 0.4$, P1-intron: $h = 7.2 \pm 0.3$, mean ± s.e.m.) was significantly higher than that of P2-specific ($h = 4.6 \pm 0.2$) and CDS ($h = 5.3 \pm 0.3$) GRFs during nc11–13 (Fig. 3d). Such difference in Hill coefficient is robust against Bcd dosage change (Supplementary Fig. 6c), suggesting that P1 activation corresponds to higher-order cooperative binding than P2 activation. Moreover, P1-specific signals exhibited a higher concentration threshold $C_0$ for activation than P2-specific and CDS signals, with their ratios being consistently larger than one (Fig. 3e). This agrees with our observation of P1 and P2 expression boundaries (Fig. 1g).

To directly quantify Bcd binding corresponding to P1 and P2 activation, we measured local enrichment of the Bcd signal in the vicinity of P1- and P2-active *hb* loci[32] (Fig. 3f; see Methods section and Supplementary Note 5). In the anterior expression domain of nc11–12 embryos, we estimated an average binding of ~6.2 Bcd molecules at P1-active loci, exceeding that of ~4.6 Bcd molecules at P2-active loci. This result confirmed that P1 activation requires cooperative binding of more Bcd molecules than P2 activation. In nc13, along with the increase in P1 activity (Fig. 1d, f), the number of bound Bcd molecules at P1-active loci dropped to ~5.4. Examining how Bcd binding at active promoter loci varied with nuclear Bcd concentration (or nuclear position)

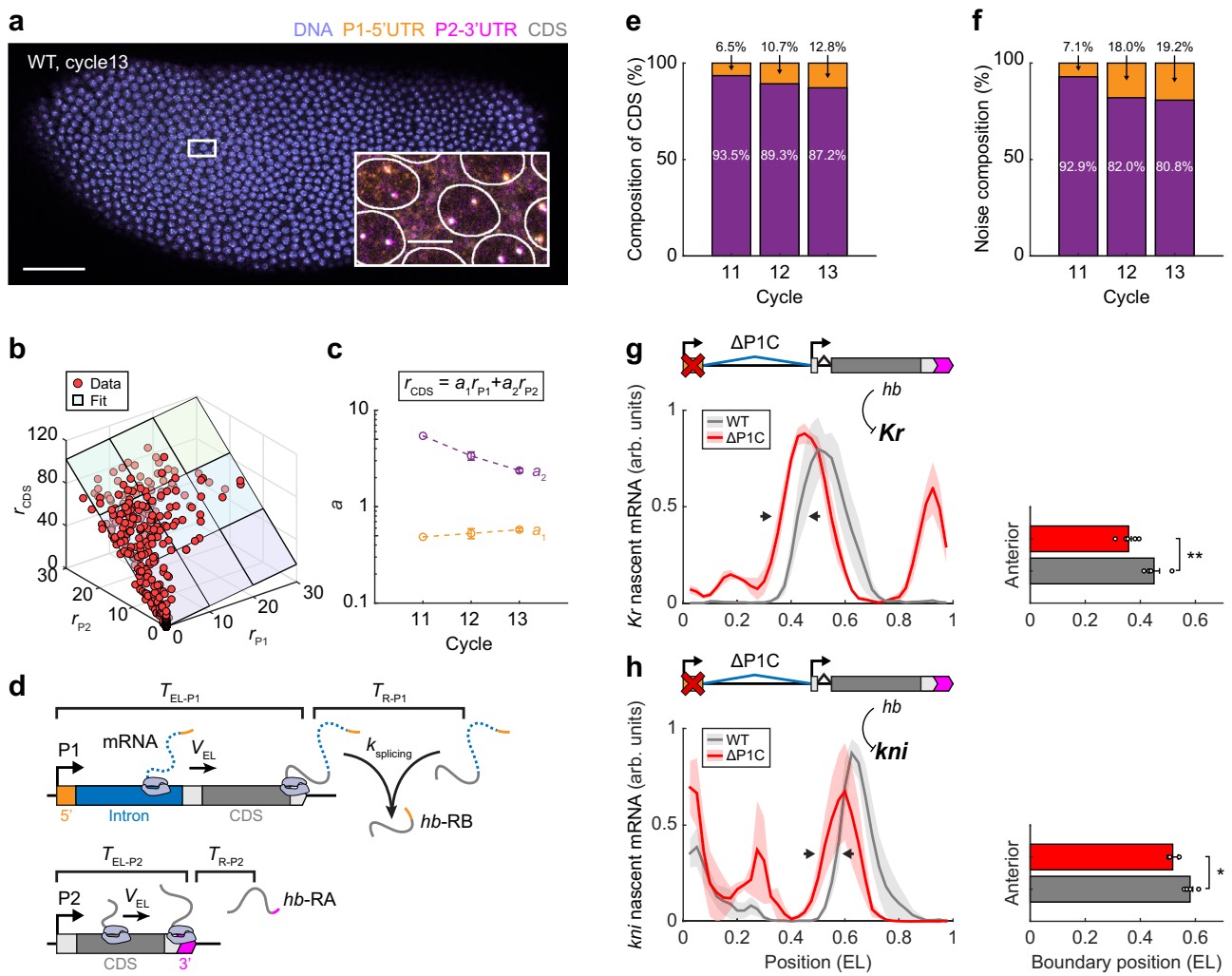

**Fig. 2 P1 contributes a modest yet non-negligible fraction of early *hb* transcription and function. a** Confocal image of a WT *Drosophila* embryo labeled for P1–5′UTR, P2–3′UTR, CDS of *hb* mRNA, and DNA at nc13. Scale bar, 50 μm. Inset, magnified view of anterior nuclei. Scale bar, 5 μm. The experiment was repeated twice, independently, with similar results. **b** Nascent P1–5′UTR, P2–3′UTR, and CDS signals per nucleus (in units of the number of mRNA molecules) were plotted against each other and fitted to a linear function. Data from a single nc12 embryo. **c** Parameters of the linear fit between P1–5′UTR, P2–3′UTR, and CDS signals during nc11–13. Data are presented as mean ± s.e.m. **d** Schematic of P1 and P2 transcription. Nascent mRNAs were elongated on the gene with a constant speed and stayed on the transcription site for an extra period before being released. The P1-specific intron was spliced during or after transcription. **e, f** The contributions of P1 and P2 to the mean (**e**) and the intrinsic noise (**f**) of the nascent CDS signal in the position range of 0.2–0.4 EL during nc11–13. **g, h** Nascent mRNA signal per nucleus as a function of the AP position for *Kr* (**g**) and *kni* (**h**) in nc14. The profile was binned from the single-nucleus data (bin size: 0.05 EL, step size: 0.025 EL). Shadings indicate s.e.m. Right: the anterior boundary positions of *Kr* and *kni* expression domains. Data are presented as mean ± s.e.m. with two-sided *t*-test (*Kr*: $p = 0.0093$; *kni*: $p = 0.013$. $*p < 0.05$; $**p < 0.01$). **c, e, f** $n = 1$, 5, and 4 biologically replicates for nc11–13, respectively (see Methods section and Supplementary Fig. 5a for embryo selection). **g, h** *Kr*: $n = 4$ and 5 biologically independent embryos for WT and ΔP1C, respectively; *kni*: $n = 4$ and 3 biologically independent embryos for WT and ΔP1C, respectively. Source data are provided as a Source Data file.

revealed that both P1- and P2-specific binding curves plateaued at ~4–5 Bcd molecules, while the P1-specific binding curve exhibited an additional plateau with ~8–10 Bcd molecules in nc11–12 (Fig. 3g and Supplementary Fig. 7). As suggested previously, these plateaus may correspond to distinct Bcd binding states at *hb* enhancers[32]. The additional Bcd binding plateau for P1 implies that P1 activation may involve more Bcd-binding steps than P2 activation. This additional plateau decreased to ~6 Bcd molecules in nc13 (Supplementary Fig. 7), implying a change in Bcd binding dynamics.

**Two Bcd-dependent enhancers synergistically drive P1 activation**. Two Bcd-dependent enhancers are involved in early *hb*

regulation. To distinguish their roles in P1 and P2 activation, we used transgenic fly lines derived from a bacterial artificial chromosome (BAC) containing the *hb* gene and its regulatory sequence[35] (Fig. 4a; see Methods section). The transgenes retained the *hb* promoters, introns, and UTRs, whereas the *hb* CDS was replaced with a *yellow* reporter gene. The distal and proximal enhancers of the two transgenes were substituted, respectively, while the regulatory sequence of the third transgene was kept intact as a control.

To measure the promoter activities of the transgene, we labeled transgenic embryos with three sets of smFISH probes targeting P1 intron, *hb* CDS, and *yellow*, respectively (Fig. 4b and see Supplementary Fig. 8 for a complementary study using different probe sets). The P1 signal corresponding to the endogenous *hb*

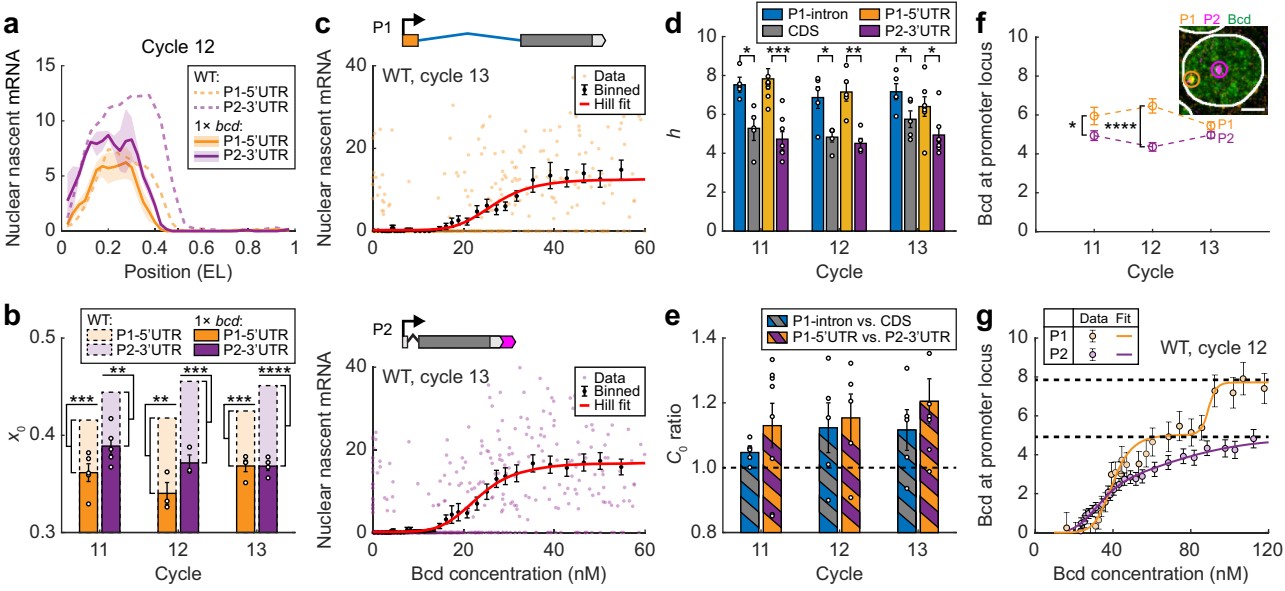

**Fig. 3 Bcd activates P1 and P2 through different regulatory relations. a** Nascent P1–5′UTR and P2–3′UTR signals per nucleus as functions of the AP position in a 1× *bcd* embryo at nc12 (solid line, bin size: 0.05 EL, step size: 0.025 EL) were compared with that in the WT (dashed line, from Fig. 1e). Shadings indicate s.e.m. **b** The boundary position of the anterior expression domain for P1–5′UTR and P2–3′UTR signals in WT and 1× *bcd* embryos during nc11–13. 1× *bcd* data are presented as mean ± s.e.m. WT: $n = 8$, 5, and 7 biologically independent embryos at nc11–13, respectively; 1× *bcd*: $n = 5$, 3, and 4 biologically independent embryos at nc11–13, respectively. Two-sided *t*-test was applied between strains (P1–5′UTR: $p = 6.7 \times 10^{-4}$, $2.6 \times 10^{-3}$, and $3.8 \times 10^{-4}$ for nc11–13, respectively; P2–3′UTR: $p = 4.3 \times 10^{-3}$, $8.2 \times 10^{-4}$, and $2.6 \times 10^{-5}$ for nc11–13, respectively. **$p < 0.01$; ***$p < 0.001$; ****$p < 0.0001$). **c** Nascent P1–5′UTR (upper) and P2–3′UTR (lower) signals in individual nuclei of a single embryo were plotted against nuclear Bcd concentration (single WT embryo, >1200 nuclei, 0.25–0.75 EL). The single-nucleus data were binned along the Bcd axis (mean ± s.e.m., bin size: 16 nM, step size: 2 nM) and fitted to Hill functions. **d** The Hill coefficient of the gene regulation function for different probe signals during nc11–13, with two-sided *t*-test (P1-Intron vs. CDS: $p = 0.011$, 0.010, and 0.046 for nc11–13, respectively; P1–5′UTR vs. P2–3′UTR: $p = 2.8 \times 10^{-4}$, $1.1 \times 10^{-3}$, and 0.041 for nc11–13, respectively. *$p < 0.05$; **$p < 0.01$; ***$p < 0.001$). **e** The ratios of the concentration threshold between P1-intron and CDS signals, and between P1–5′UTR and P2–3′UTR signals, during nc11–13. **f** The average number of Bcd molecules bound at P1- and P2-active *hb* loci in the anterior expression domain (0.2–0.35 EL) during nc11–13, with two-sided *t*-test ($p = 0.039$, $2.9 \times 10^{-7}$, and 0.067 for nc11–13, respectively. *$p < 0.05$; ****$p < 0.0001$). Inset: an anterior nucleus labeled for P1–5′UTR and P2–3′UTR of *hb* mRNAs and Bcd protein. The enriched Bcd signal in the vicinity of the promoter (yellow and purple circles) was measured. Scale bar, 2 μm. **g** Bcd binding at P1- and P2-active *hb* loci as a function of nuclear Bcd concentration at nc12. Data were pooled from $n = 5$ biologically independent embryos and were binned by nuclear AP position (bin size: 0.035 EL, step size: 0.01 EL). The binned data were fitted to multi-Hill functions. Dashed lines highlight discrete binding plateaus for each promoter. **d–f** Data are presented as mean ± s.e.m. P1–5′UTR and P2–3′UTR: $n = 8$, 5, and 7 biologically independent embryos at nc11–13, respectively; P1-Intron and CDS: $n = 5$ biologically independent embryos at each nuclear cycle. Source data are provided as a Source Data file.

gene was identified and excluded based on its colocalization with the *hb* CDS signal. We compared the anterior P1 and *yellow* (mainly from P2) signals in the control and enhancer-removed transgenes in nc11–13 embryos (Fig. 4c and Supplementary Fig. 9a, b). Removal of either enhancer significantly lowered the percentage of the *yellow*-positive anterior nuclei by ~10% in nc11–12 (Fig. 4d), consistent with previous reports that both enhancers are required for authentic *hb* expression[23,30]. Similarly, removing either enhancer lowered the percentage of P1-active anterior nuclei by >33% in nc11–12 embryos (Fig. 4d), suggesting that early P1 activation also relies on both enhancers.

Comparing the *yellow* expression profiles between the control and enhancer-removed transgenes (Fig. 4e and Supplementary Fig. 9c, d) revealed a decrease in the maximum anterior expression level $r_{max}$ by ~30–40% in response to proximal or distal enhancer removal in nc11–12 (Fig. 4f). This agrees with a previous report that P2 subadditively integrates regulatory inputs from different enhancers via enhancer competition[23]. In contrast, removing the proximal and distal enhancers decreased the maximum expression level $r_{max}$ of P1 by ≥60% and ≥37% in nc11–12, respectively (Fig. 4f). It suggests that the two enhancers activate P1 with little competition.

Besides affecting the expression amplitude, both enhancers are critical for promoter expression boundaries. Specifically, removing

the proximal enhancer shifted the *yellow* expression boundary towards the anterior pole, while removing the distal enhancer caused a posterior shift of the *yellow* profile (Fig. 4g). These results agree with the enhancer competition model for P2 activation[23], in which the expression boundary resulting from two enhancers lies between that from individual ones. In contrast, the deletion of either enhancer caused an anterior shift of the P1 expression boundary (Fig. 4h), which is inconsistent with the enhancer competition model. It shows that the existence of a second enhancer helps activate P1 at lower Bcd concentrations. Thus, the two enhancers may interact synergistically to drive P1 activation.

Moreover, the Hill coefficient of P1 dropped significantly from ~6.6 to ~4.4, a value close to the Hill coefficient of P2, in response to enhancer removal (Supplementary Fig. 9e). It suggests that synergistic action of the two enhancers involves Bcd binding at both of them. In contrast, the Hill coefficient of P2 does not change significantly with enhancer removal, indicating Bcd binding at a single enhancer.

**Promoter-specific transcription kinetics reveal a unified scheme of enhancer-promoter interaction.** Nascent mRNA copy number statistics reflect the microscopic mechanisms of gene regulation[56]. Previous studies reported a super-Poissonian distribution of the nascent mRNA copy number on individual *hb*

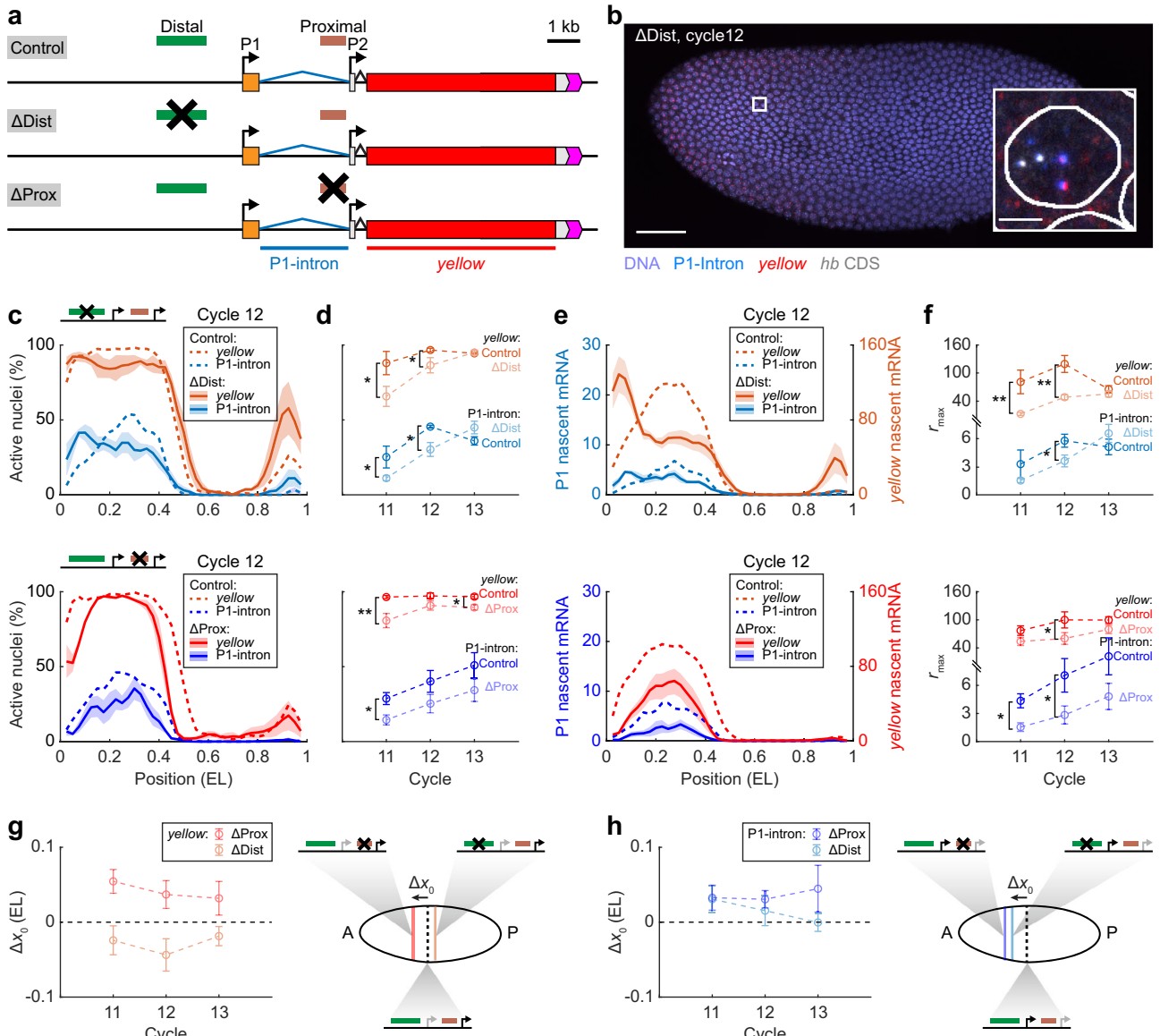

**Fig. 4 Two enhancers combine differently to drive P1 and P2 activation. a** Schematic of *hb* reporter constructs with enhancer replacements. A construct without enhancer deletion was used as a control. Two smFISH probe sets were used to label different regions of reporter mRNAs: blue, P1-intron probes; red, *yellow* probes. **b** Confocal image of a distal-enhancer-removed embryo labeled for P1-intron, *yellow*, and *hb* CDS at nc12. Scale bar, 50 μm. Inset, magnified view of a single anterior nucleus. Scale bar, 2 μm. The experiment was repeated twice, independently, with similar results. **c** Percentage of active nuclei as a function of the AP position for P1-intron and *yellow* signals in nc12 embryos of different constructs. Shadings indicate s.e.m. **d** Average percentage of active nuclei in the position range of 0.2–0.4 EL for P1-intron and *yellow* signals in different constructs during nc11–13, with two-sided *t*-test (ΔDist vs. Control: $p = 0.048$, 0.044, and 0.73 for *yellow* in nc11–13, respectively; $p = 0.049$, 0.018, and 0.13 for P1-intron in nc11–13, respectively. ΔProx vs. Control: $p = 0.0035$, 0.13, and 0.025 for *yellow* in nc11–13, respectively; $p = 0.027$, 0.17, and 0.18 for P1-intron in nc11–13, respectively. *$p < 0.05$; **$p < 0.01$). **e** Nascent P1-intron and *yellow* signals per nucleus as functions of the AP position in nc12 embryos of different constructs. Shadings indicate s.e.m. **f** The maximal signal level of the anterior expression domain for P1-intron and *yellow* signals in different constructs during nc11–13, with two-sided *t*-test (ΔDist vs. Control: $p = 0.0099$, 0.0091, and 0.36 for *yellow* in nc11–13, respectively; $p = 0.19$, 0.045, and 0.30 for P1-intron in nc11–13, respectively. ΔProx vs. Control: $p = 0.13$, 0.044, and 0.084 for *yellow* in nc11–13, respectively; $p = 0.016$, 0.033, and 0.14 for P1-intron in nc11–13, respectively. *$p < 0.05$; **$p < 0.01$). **g, h** The average boundary shift of the anterior expression domain for *yellow* (**g**) and P1-intron (**h**) signals upon removing one enhancer. Error bars were computed from the standard errors of boundary positions for enhancer-deleted and control lines using error transfer formula. Right: schematic of boundary shift. **c–h** Data are presented as mean ± s.e.m. ΔDist: $n = 6$, 4, and 4 biologically independent embryos at nc11–13, respectively; ΔDist control: $n = 4$ biologically independent embryos at each nuclear cycle. ΔProx: $n = 5$, 4, and 4 biologically independent embryos at nc11–13, respectively; ΔProx control: $n = 7$, 4, and 6 biologically independent embryos at nc11–13, respectively. The spatial profile of each embryo was binned from the single-nucleus data (bin size: 0.05 EL, step size: 0.025 EL). Source data are provided as a Source Data file.

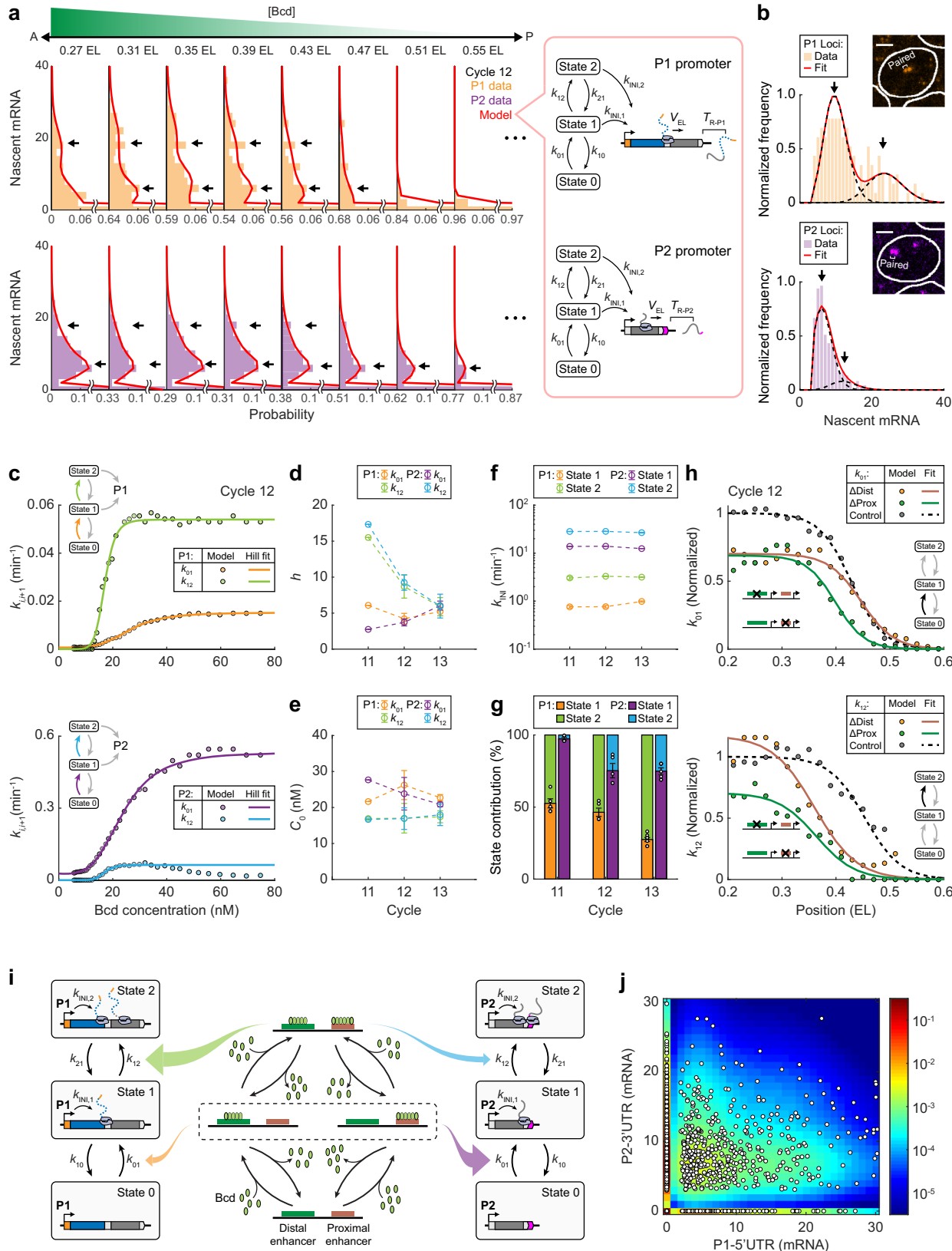

gene loci[31,32,45]. Such distribution can be explained by a minimal model of two-state transcription kinetics[42,46,56,57] with Bcd modulating the frequency of stochastic transition from an inactive to an active transcription state[32].

To uncover the kinetics of P1 and P2 transcription, we measured, for each embryo, the distributions of P1 and P2

nascent mRNA signals in different Bcd concentration ranges (Fig. 5a). Contrary to the prediction from the two-state model[46], each promoter exhibited a wide distribution with more than one population of active loci characterized by different expression levels (Fig. 5a). A natural explanation of this phenomenon is that some of the observed promoter loci may indeed be a pair of

**Fig. 5 Three-state promoter kinetics reveal a unified scheme of P1 and P2 regulation. a** Histograms of nascent P1–5′UTR (upper) and P2–3′UTR (lower) signals at individual *hb* gene loci in different position ranges (single embryo, the center of each position range is indicated above the histogram). Each histogram was fitted to a three-state transcription model (right). **b** Histograms of nascent P1–5′UTR (upper) and P2–3′UTR (lower) signals at active sister loci in the position range of 0.2–0.4 EL. Each histogram was fitted to two Poisson distributions. Data pooled from seven embryos at nc13. Insets, images of a single anterior nucleus with active sister loci pairs. **c** Promoter activation rates for P1 and P2 estimated from five embryos at nc12 (binned along the AP axis, bin size: 0.1 EL, step size: 0.01 EL) were plotted against nuclear Bcd concentration and fitted to Hill functions. **d, e** The Hill coefficients (**d**) and concentration thresholds (**e**) of promoter activation rates for P1 and P2 during nc11–13. **f** The transcription initiation rates of the two active states for P1 and P2 during nc11–13. **g** The contributions of states 1 and 2 to P1 and P2 transcription in the position range of 0.2–0.4 EL during nc11–13. **h** P1 activation rates as functions of the AP position for different constructs at nc12 (data from 4 embryos for each reporter construct, bin size: 0.1 EL, step size: 0.02 EL). Data were normalized and compared between the distal- or proximal-enhancer-removed constructs and the control. **i** Schematic of P1 and P2 regulation by cooperative Bcd binding at the proximal and distal enhancers. **j** The joint distribution of nascent P1–5′UTR and P2–3′UTR signals at individual *hb* gene loci in the position range of 0.2–0.4 EL compared with model prediction. Data pooled from 17 embryos during nc11–13. White circles, individual loci. Color code, probability estimated from a unified model of P1 and P2 transcription. **d–g** Data are presented as mean ± s.e.m. $n = 5$, 5, and 7 biologically independent embryos at nc11–13, respectively. Source data are provided as a Source Data file.

closely located sister loci that are indistinguishable under the microscope[31,45]. However, the nascent mRNA signal from individual optically resolved sister loci also exhibited two active populations, with >38% of P1 and >17% of P2 active sister loci corresponding to the minor population (Fig. 5b and see Supplementary Note 6). Based on the correspondence between distribution modality and the number of transcription states, P1 and P2 transcription should be modeled as a three-state process (see Supplementary Note 6).

In this model, a promoter randomly switches between an inactive state (state 0) and two active states (states 1 and 2) with Poissonian rates $k_{ij}$ ($i, j = 0, 1, 2$), whereas new transcripts are only initiated in active states with rates $k_{INI,i}$. Following transcription initiation, each nascent mRNA elongates with constant speed $V_{EL}$, and resides on the gene for an extra termination period $T_R$ before being released (Fig. 5a). With a detailed balance between states, we solved the steady-state distribution of nascent mRNA signal per promoter locus and compared it with experimental data to infer the kinetic parameters (Fig. 5a and see Supplementary Note 6). We found that P1 and P2 both followed a sequential activation scheme from state 0 to state 2 (Supplementary Fig. 10a and Supplementary Note 6). Bcd mainly regulated the activation rates, $k_{01}$ and $k_{12}$, while other kinetic rates remained constant (Supplementary Fig. 10b and see Supplementary Fig. 10c, d for a complementary study using the CDS signal as a proxy for P2).

For both promoters, the Bcd dependence of $k_{01}$ and $k_{12}$ satisfied Hill functions (Fig. 5c), with the corresponding Hill coefficient $h$ and concentration threshold $C_0$ identical between the promoters (Fig. 5d, e). Specifically, the Hill coefficient of $k_{01}$ was ~4.5 (P1: $h = 5.2 \pm 0.5$, P2: $h = 4.2 \pm 1.0$, Fig. 5d), close to that of P2-GRF. In contrast, the Hill coefficient of $k_{12}$ was ~10 (P1: $h = 10.0 \pm 2.8$, P2: $h = 10.8 \pm 3.3$, Fig. 5d). $h$ of $k_{01}$ and $k_{12}$ each matched a plateau in the Bcd binding curves of *hb* promoters (Fig. 3g), suggesting that each promoter activation step may involve a cooperative Bcd binding event. The concentration threshold of $k_{01}$ was ~24 nM (P1: $C_0 = 23.5 \pm 1.4$, P2: $C_0 = 24.1 \pm 2.0$, Fig. 5e), which was close to that of P2-GRF. In contrast, the concentration threshold of $k_{12}$ was only ~17 nM (P1: $C_0 = 17.1 \pm 0.2$, P2: $C_0 = 17.2 \pm 0.4$, Fig. 5e), suggesting that the second Bcd-binding event is easier to happen than the first one.

In addition to the Bcd dependence, the ratio between the activation and inactivation rates determines the probability of each transcription state. In the anterior expression domain, $k_{01}/k_{10}$ for P1 was smaller than that for P2 by ~4–7 folds (Supplementary Fig. 10e), indicating that P2 is more likely to be activated than P1. In contrast, $k_{12}/k_{21}$ for P1 exceeded that for P2 by >3 folds (Supplementary Fig. 10f), indicating that active P1 is more inclined to reach state 2 than active P2. For both

promoters, the transcription initiation rate of state 2 was larger than that of state 1 (Fig. 5f). Therefore, state 2 contributed much more to P1 transcription than to P2 transcription (57.9% vs. 17.5%, Fig. 5g). This explains the difference (in $h$ and $C_0$) between the P1- and P2-GRFs. I.e., the differential regulation of P1 and P2 results from their preference for different active states.

To relate transcription kinetics with enhancer activities, we applied theoretical analysis to enhancer deletion experiments. For early P1 transcription (nc11–12), the removal of either enhancer yielded a uniform decrease of $k_{01}$ by ~30% in the anterior expression domain (Fig. 5h and Supplementary Fig. 10g). Deleting the proximal enhancer also caused a modest anterior shift of the $k_{01}$ boundary (Fig. 5h). These results suggest that either enhancer can drive promoter activation to state 1. In contrast, deleting either enhancer caused a dramatic anterior shift of the $k_{12}$ boundary (Fig. 5h and Supplementary Fig. 10h). This indicates that state 2 is primarily driven by the synergistic action of both enhancers. When the Bcd concentration is sufficiently high, state 2 may also be driven by a single enhancer, possibly due to extra Bcd binding at weak binding sites. These mechanisms explain the difference between P1 and P2 in the expression boundary shift upon enhancer removal (Fig. 4g, h).

Altogether, we propose that the two *hb* promoters follow a common kinetic scheme of Bcd-dependent activation (Fig. 5i and see Supplementary Note 6). Cooperative Bcd binding at either enhancer can drive a promoter to a basal active state, while full-power transcription requires additional Bcd binding at the second enhancer. The two promoters differ in their responses to different Bcd binding configurations. P2 is primarily activated by Bcd binding at a single enhancer, which results in competitive action of the two enhancers. In contrast, P1 activation relies more on Bcd binding at both enhancers, which leads to synergistic enhancer action. The predicted joint distribution of P1 and P2 activities agreed well with experimental results (Fig. 5j and see Supplementary Note 6).

## Discussion

Traditional FISH and *lacZ* reporter experiments reported that Bcd-dependent *hb* transcription in early embryogenesis only involves the P2 promoter[33,34,36,37], while P1 is Bcd-insensitive and remains silent until late nc14[27,30,36]. Here, using smFISH method with single-molecule sensitivity, we showed that Bcd activated both promoters in the anterior domain of nc11–13 embryos. P1 contributes up to ~13% of nascent *hb* signal and up to ~20% of nascent *hb* noise. These fractions are modest compared to those of P2, yet significant enough to affect the expression patterns of *hb*-target genes. Thus, the Bcd-dependent P1 expression is not negligible.

According to previous studies, the difference in P1 and P2 expression levels is due to specific motif codes in promoter sequences. P2 contains Zelda binding sites and a strong TATA box to facilitate chromatin opening and promoter activation[16,30,58]. In contrast, the lack of Zelda binding sites and TATA box in the P1 promoter significantly impedes local chromatin opening and promoter activation[17,38,59]. However, our results suggest that the lack of these motif codes may be insufficient to completely block early P1 activity. In fact, the local chromatin state of a gene has been reported to be highly dynamic and can randomly switch between "open" and "close" to allow transient gene expression[13,60]. This effect has been proposed as a mechanism for transcriptional bursting[60–62], consistent with our observation of bursty P1 transcription. Moreover, our observation that the percentage of active P1 loci increased with the nuclear cycle suggests a gradual increase in chromatin opening frequency during development. The stronger P1 activity in late nc14 may be a continuation of this trend.

In addition to the expression amplitude, the Bcd dependence of P1 and P2 showed quantitative differences. P1 activation requires a higher Bcd concentration and binding of more Bcd molecules than P2 activation. One possible mechanism for this difference is that the two promoters are driven by different enhancers. However, we showed that early P1 and P2 activation relied on the same pair of enhancers[35]. Another possibility is that the two promoters distinguish their behaviors by mutual interaction[63]. However, we found little correlation between P1 and P2 activities. This is consistent with a previous promoter deletion experiment[30] and indicates no interaction between the two promoters. Thus, the difference in Bcd dependence originates from complex enhancer-promoter interactions.

One form of complex enhancer-promoter interaction is the competitive action of the proximal and shadow enhancers for *hb* P2 activation[23]. Here, each enhancer corresponds to a unique Bcd-dependent regulatory effect, and the two enhancers compete for promoter activation[12]. The resulting P2 expression is a sub-additive integration of individual enhancers' regulatory effect. This mechanism can lead to a different Bcd-dependent relationship if the enhancers' relative weights in the integration change. However, the P1 expression boundary always shifts anteriorly in response to the deletion of either enhancer, suggesting a synergistic, as opposed to competitive, action of the two enhancers for P1 activation. This result also revealed that the Bcd dependence of P1 and P2 differed even in the case of a single enhancer, indicating that the TF dependence of an enhancer's regulatory effect may be promoter-specific.

To understand the mechanism of promoter-specific interactions with enhancers, we analyzed the transcription kinetics of individual promoters. Unlike the previous model of two-state *hb* transcription[31,32,45], we found that both P1 and P2 activities need to be described by three-state kinetics with sequential activation steps. Multi-step transcriptional activation was previously proposed to model enhancer-, promoter-, or chromatin-related intermediate states during activation[13,60,62,64,65]. However, these intermediate states are rarely experimentally detectable. In the case of *hb*, previous measurements mixed signals from P1 and P2, whose difference easily overwhelmed the subtle signatures of different transcription states. Here, the identification of multiple active states was possible owing to promoter-specific demixing of the *hb* nascent mRNA signal. Thus, it is important to distinguish alternative promoters in the study of complex gene regulation.

Our model revealed that Bcd regulates both activation steps ($k_{01}$ and $k_{12}$) through cooperative binding. This suggests that the two active states of a promoter may correspond to different Bcd binding states at *hb*. Previous literature has shown evidence of multiple Bcd binding states at *hb*[32]. However, only one binding state (with ~4–5 Bcd molecules) was reported to coincide with gene activation[32]. The biological functions of the other binding states were unclear. Here, we showed that the binding of ~4–5 Bcd molecules happened at a single enhancer (either proximal or distal). Enhancers at this binding state can competitively drive promoter activation, consistent with a previous study of *hb* P2 activation[23]. In contrast, the binding of ~8–10 Bcd molecules involved both enhancers, leading to synergistic activation of a promoter. Such a mechanism was proposed for some eukaryotic genes[23,66] but has never been reported for *hb*.

P1 and P2 differ in their preference for different Bcd binding states. Specifically, the P1 response to the lower Bcd binding state is much less than that of P2. A possible reason is that P1 lacks specific motif codes for chromatin opening[30]. Thus, its activation may require TF binding at enhancers to help open the local chromatin configuration[61,62]. Binding of more Bcd molecules at both enhancers may be more effective for chromatin opening. However, since P1 locates between the two enhancers, binding of Bcd (and other regulatory factors) at the proximal enhancer may physically block P1 transcription. It is unclear how P1 coordinates its activation and transcription. One possibility is that TF binding at enhancers is only transiently needed to form the preinitiation complex. Future experiments using high-resolution live imaging techniques would likely solve this puzzle[67,68].

Altogether, these results showed that a single kinetic scheme could create apparently different types of enhancer-promoter interactions. Such a unified scheme may be shared by alternative promoter regulation in many eukaryotic systems and may be crucial for phenotypic complexity in higher eukaryotes[7]. Moreover, our combined experimental and theoretical approach directly relates TF binding at individual enhancers with the stochastic activity of each promoter. A generalization of this approach (e.g., by including more regulatory factors) will help understand the regulatory mechanisms for cell fate decisions and enable the precise design of synthetic gene circuits[69].

## Methods

**Fly strains**. Oregon-R (OreR) strain was used as the wild type. CRISPR mutant strains with P1 and P2 deletions (ΔP1C and ΔP2C) were developed previously[30] and were obtained as gifts from Dr. Stephen Small (New York University) and Dr. Pinar Onal (Northwestern University). 1×*bcd* strain (+/CyO-bcd + ; E1s) was developed previously[51] and was obtained as a gift from Dr. Jun Ma (Zhejiang University). *hb*-BAC reporter constructs were previously developed[35]. The distal-enhancer-removed BAC construct and its control were obtained as gifts from Dr. Michael Perry (University of California San Diego) and Dr. Alistair Boettiger (Stanford University). The proximal-enhancer-removed construct and its control were rebuilt as previously described[35]. Transgenes in these constructs were constructed from BAC CH322–55J23, which contains a 20-kbp *Drosophila* genomic sequence encompassing the *hb* gene and its proximal and distal enhancers. A *yellow-kanamycin* fusion was used to replace the *hb* CDS in all BACs, while the proximal or distal enhancers in the BACs were substituted with *ampicillin*. All BACs were integrated on chromosome 2 of *Drosophila*. The distal-enhancer-removed transgene and its control were integrated into landing site VK37 (Bloomington Stock Center number 24872). The proximal-enhancer-removed transgene and its control were integrated into landing site attP40 (Bloomington Stock Center number 25709).

**smFISH probe design**. Sets of DNA oligonucleotides complementary to the target transcripts (nine probes for *hb* P1 5′UTR, 32 probes for *hb* P1 intron, 48 probes for *hb* CDS, eight probes for *hb* P2 3′UTR, 48 probes for *yellow*, 33 probes for *Kr*, and 29 probes for *kni*) were designed (Supplementary Table 1) and synthesized (Biosearch Technologies). Each probe was ordered with a 3′ amine group (mdC(TEG-Amino)) and was conjugated to various fluorophores[32]. In most experiments, *hb* P1–5′UTR and intron probes were conjugated with tetramethylrhodamine (TAMRA; Thermo Fisher Scientific, C6123). *hb* P2–3′UTR, *yellow*, and *Kr* probes were conjugated with Alexa Fluor™ 647 (Invitrogen, A20106). *kni* probes were conjugated with Alexa Fluor™ 488 (Invitrogen, A20100). *hb* CDS probes were conjugated with either Alexa Fluor™ 647 (Invitrogen, A20106) or Alexa Fluor™ 488 (Invitrogen, A20100). In a control experiment, P1–5′UTR probes with Alexa Fluor™ 647 and P2–3′UTR probes with TAMRA were used to label WT embryos. A comparison between signals from the same probe set with different fluorophores

revealed that the efficiencies of different fluorescent detectors were comparable (Supplementary Fig. 2e, f).

**Embryo staining**. Embryo collection, fixation, and labeling were performed according to a previously published protocol[32]. Briefly, 2-h-old embryos were collected at 25 °C, fixed with 4% paraformaldehyde solution, and stored in 100% methanol at −20 °C. For smFISH, fixed embryos were rehydrated (4 × 10 min) in PBTx (1× PBS, 0.1% (v/v) Triton X-100) at room temperature, washed (2 × 10 min) in hybridization wash buffer at 30 °C, and incubated with the probe-containing hybridization buffer (2× SSC, 20% (w/v) formamide, 0.1% (v/v) Triton X-100) at 30 °C overnight. After hybridization, embryos were washed in hybridization wash buffer at 30 °C (2 × 10 min) and in 2× SSC at room temperature (2 × 10 min). For immunofluorescence (IF), embryos were washed (4 × 10 min) in PBTx and blocked in PBT-B (1× PBS, 20% (v/v) western blocking reagent (Roche, 11921673001), 2 mM ribonucleoside vanadyl complex (NEB, S1402S), 0.1% (v/v) Triton X-100) at room temperature for 1 h. The preabsorbed rabbit anti-Bcd primary antibody (Santa Cruz Biotechnology, SC-66818, 1:50 (v/v) dilution in PBT-B) was incubated with embryos at 4 °C for 20 h. Following primary antibody staining, embryos were further washed (4 × 10 min) in PBTx, blocked in PBT-B at room temperature for 1 h, and incubated with goat anti-rabbit IgG secondary antibody conjugated with Alexa Fluor™ 488 (Invitrogen, A11034, 1:500 (v/v) dilution in PBT-B) at room temperature for 1 h. For DNA counterstaining, embryos were washed (4 × 10 min) in PBTx and stained with Hoechst 33342 at room temperature for 10 min. Following additional washes (4 × 10 min) in PBTx, embryos were mounted in Aqua-Poly/Mount (Polysciences, 18606). Imaging was performed after the samples were completely solidified.

**HCR-FISH probe design and labeling**. HCR-FISH was used as a replacement of smFISH to identify active *yellow* or *lacZ* loci in some control experiments (Supplementary Figs. 4 and 9). Design of HCR-FISH probes and labeling of embryos are described in detail in Supplementary Note 2.

**Imaging**. Most embryos were imaged using a Zeiss LSM 880 laser scanning confocal microscope equipped with a GaAsP detector and a 63× oil-immersion objective (1.4 NA). 16-bit image stacks were acquired with a pixel size of $71 \times 71$ nm$^2$ and a z-step size of 0.32 μm. A small number of embryos ($n = 5$) were imaged using a Leica TCS SP8 confocal microscope equipped with a GaAsP detector and a 63× oil-immersion objective (1.4 NA). 12-bit image stacks were acquired with a pixel size of $81 \times 81$ nm$^2$ and a z-step size of 0.3 μm. To image *hb* transcription, nc11–13 embryos at the mitotic interphase were selected based on the number and shape of the nuclei (Hoechst signal). Pre-gastrulation nc14 embryos were selected to image *Kr* and *kni* activities. Approximately 10 μm of the cortex layer of each embryo was imaged. For promoter mutant lines, each mutant allele was made heterozygous with a TM3 Sb balancer containing a *hb-lacZ* reporter. To identify homozygous mutant embryos, the *lacZ* FISH signal (labeled by Alexa Fluor™ 594) was imaged first, and those embryos without active *lacZ* loci were selected for subsequent imaging.

**Preprocessing and nuclear segmentation**. Image processing and data analysis followed a previously developed pipeline[32] with updated algorithms to improve accuracy and efficiency. Briefly, raw images were divided by a normalized flat-field image to correct for monochromatic aberrations. Three-dimensional (3D) segmentation of nuclei from the Hoechst image stack was done using a combination of local threshold (to optimize a circularity parameter) and watershed (to separate the merged nuclei). The nuclear cleavage cycle of the embryo was determined by the number of recognized nuclei. Embryo boundary was identified from the averaged Hoechst image by thresholding image pixels outside the nuclear area. This boundary was then used to determine the AP position of each nucleus.

**mRNA quantification**. Spot candidates in smFISH images were identified as 3D local maxima in the image stack. Since the splicing process occurs inside the nucleus, intron spot candidates were only identified in the nuclear region. The local intensity profile of each candidate was fitted to a two-dimensional (2D) Gaussian function to extract the peak height ($I_{peak}$) and radius ($\sigma_0$). The spot intensity was calculated as $I = 2\pi I_{peak}\sigma_0^2$. By comparing the joint distribution of peak height and radius between the anterior and posterior spots, a 2D threshold was determined to distinguish real mRNA spots from background noise (Supplementary Fig. 2a). The typical intensity, $I_0$, of a single mRNA molecule was extracted by fitting the primary peak of the spot intensity distribution to a multi-Gaussian function (Supplementary Fig. 2b). A threshold of $3I_0$ was defined to identify sites of active transcription from mRNA spots inside each nucleus. The equivalent number of nascent transcripts at each transcription site was estimated by dividing the intensity of the transcription site by $I_0$.

To identify signals corresponding to the same *hb* locus in two different smFISH channels, we calculated the mutual distance between every possible pair of active transcription sites detected in different smFISH channels. The distribution of all the mutual distances in an embryo exhibited a clear population within a small distance range (<0.55 μm), outside which the distribution became flat (Supplementary Fig. 2c). This population corresponded to colocalized pairs of active transcription sites. Using a threshold distance of 0.55 μm, we identified these colocalized pairs, each of which belonged to a single *hb* locus. In contrast, each unpaired active transcription site belonged to a different *hb* locus, whose signal in the other smFISH channel was zero.

In some anterior nuclei, replication of the *hb* gene can lead to >2 bright FISH spots[31]. To identify nascent mRNA signals corresponding to newly replicated sister loci pairs, we calculated the mutual distance between every possible pair of active transcription sites in each nucleus. The distribution of all the mutual distances in the embryo exhibited two distinct populations (Supplementary Fig. 2d). The population with smaller mutual distances corresponds to sister loci pairs, while the other population corresponds to unpaired homologous loci. Using a threshold distance of 0.71 μm that lay at the valley point between the two populations in the distribution, sister loci pairs were distinguished from unpaired homologous loci.

Nuclei with and without paired active loci were both analyzed in the paper. For single-locus analysis, we roughly estimated the number of silent loci in each nucleus using a default criterion. I.e., nuclei with paired active loci are post-replication and have four copies of the *hb* gene, while nuclei without paired active loci were coarsely assumed to be pre-replication and have two copies of the *hb* gene. To avoid the possible inaccuracy in gene copy inference, we summed over the nascent signal in each nucleus as an alternative measure of promoter activity for mean-level analysis.

**Protein quantification**. The average immunofluorescence (IF) intensity of each nucleus was calculated from the central z-slice of the nucleus. IF spots in the cytoplasm were identified and quantified to determine the typical intensity, $I_1$, of a single protein molecule, following the same procedure used for the smFISH signal (Supplementary Fig. 6a). The absolute protein concentration of a nucleus was estimated by dividing the average IF intensity of the nucleus by $(2\pi)^{1/2}\sigma_z I_1$, where $\sigma_z$ is the half-width of the single-protein intensity profile in the z dimension.

In the preceding steps, nuclear segmentation, embryo boundary identification, and detection of active transcription sites could be further refined and corrected manually using custom MATLAB graphical user interfaces.

**Measuring the spatial profile of promoter activity**. We analyzed the expression profile of a promoter using embryos in mid-to-late mitotic interphase to ensure steady-state promoter activity. For each embryo, we plotted the nascent mRNA signal ($r$) against nuclear position ($x$) for all nuclei and binned individual data points by $x$. Within the range of 0.25–0.75 EL, we used a least-squares algorithm (the "nlinfit" function in MATLAB) to fit the binned data to a logistic function:

$$r = r_{\max} \frac{e^{-(x-x_0)/d}}{e^{-(x-x_0)/d} + 1} + r_0 \tag{3}$$

where $x_0$ is the boundary position of the expression domain, $d$ is the half-width of the transition region, $r_{\max}$ is the maximal transcription level induced in the anterior region, and $r_0$ denotes the basal activity in the posterior part.

**Measuring the fluctuation of promoter activity**. At the single-molecule level, gene expression constantly varies over time and between different cells. Based on the correlation between the homologous loci in the same cell, the fluctuation or noise of gene expression may be divided into two parts: the intrinsic noise due to the inherent stochasticity of biochemical reactions and the extrinsic noise caused by cell-to-cell variability of the microenvironment[70].

To characterize the expression variability of different *hb* promoters, we computed, for each promoter, the Fano factor ($F$) of nascent mRNA signal ($r$, in units of the number of molecules) per locus in the anterior expression domain (0.2–0.4 EL)[32]:

$$F = \frac{\sigma^2}{\langle r \rangle} \tag{4}$$

where $\langle r \rangle$ and $\sigma$ are the mean and standard deviation of the single-locus data, respectively.

We quantified the intrinsic noise of P1 and P2 expression in the anterior expression domain (0.2–0.4 EL) of an embryo using the following formula[44]:

$$\eta^2 = \frac{\langle (m_1 - m_2)^2 \rangle}{2\langle m_1 \rangle \langle m_2 \rangle} \tag{5}$$

where $m_1$ and $m_2$ are nascent mRNA signals at two homologous loci in the same nucleus measured using a given probe set, respectively.

**Measuring the correlation between promoter loci**. To distinguish the intrinsic and extrinsic noise of different promoter activities, we computed the correlation coefficient ($\rho$) of the nascent mRNA signal between the two homologous copies of a given promoter within the same nucleus. Specifically, we divided the single-locus data of the promoter activity (in units of the number of mRNA molecules) in an embryo into two groups, $r_1$ and $r_2$. Each group corresponded to one of the two homologous loci in the nucleus. In a given region of the embryo, we applied the

following formula:

$$\rho = \frac{\langle(r_1 - \langle r_1 \rangle) \cdot (r_2 - \langle r_2 \rangle)\rangle}{\sigma_1 \sigma_2} \tag{6}$$

where $\langle r_1 \rangle$, $\langle r_2 \rangle$, $\sigma_1$, and $\sigma_2$ are the mean and standard deviation of each group, respectively. In the anterior expression domain (0.2–0.4 EL), P1, P2, and CDS signals showed little correlation (<0.15; Supplementary Fig. 3c), agreeing with previous reports of loci independence[31,32]. Thus, intrinsic noise dominates P1 and P2 expression. To further evaluate the interaction between the two promoters, we computed the correlation coefficient between P1 and P2 signals from the same (intra-allele) or different (inter-allele) *hb* loci in the nucleus using the same formula, where $r_1$ and $r_2$ denote the activities of individual P1 and P2 copies, respectively. Both quantities were at low levels (<0.06; Supplementary Fig. 3d), indicating that the two promoters do not interact during expression.

**Estimating promoter contributions to *hb* activity.** To estimate the contributions of P1 and P2 activities to nascent *hb* transcription, we co-labeled fly embryos with P1–5′UTR, P2–3′UTR, and CDS probes. For each nucleus in the position range of 0.3–0.6 EL of an embryo, we computed the nascent signal of each probe set (in units of the number of mRNA molecules). By locally averaging the signal over the nearest five neighboring nuclei, we plotted the three FISH signals against each other and fitted the data to Eq. (1) using linear regression. $a_1$ extracted from most embryos at mitotic interphase exhibited a typical value of ~0.5 (Supplementary Fig. 5a), close to the ratio between CDS and P1–5′UTR signals in ΔP2C mutant (Supplementary Fig. 4a, b; see Supplementary Note 3). In contrast, embryos close to mitosis gave $a_1 \gtrsim 1$, as transcription initiation has been turned off at that moment (Supplementary Fig. 5a). These embryos were excluded from analysis using a threshold $a_1$ of 0.8.

With the inferred ratio parameters $a_1$ and $a_2$, we scaled nuclear P1–5′UTR and P2–3′UTR signals to compare with nuclear CDS signal in different parts of the embryo (Supplementary Fig. 5c). The two scaled signals reconstructed most of the CDS signal in the anterior expression domain (0.2–0.4 EL). We computed the average P1 contribution to the anterior nascent CDS signal as

$$f_{P1-CDS} = \frac{a_1 r_{P1}}{r_{CDS}} \tag{7}$$

where $r_{CDS}$ and $r_{P1}$ are the average nascent CDS and P1–5′UTR signals per nucleus, respectively. The rest of the nascent CDS signal came from P2. Extra CDS component existed in terminal regions (0–0.2 and 0.8–1 EL; Supplementary Fig. 5c), suggesting different ratio parameters for these regions.

To compute promoter contributions to the intrinsic noise of nascent *hb* transcription, we estimated $\eta^2_{P1-CDS}$ in Eq. (3) from the intrinsic noise of the nascent P1–5′UTR signal (see Supplementary Note 6). The contribution of P1 to the intrinsic noise of nascent CDS signal was calculated as:

$$w_{P1-CDS} = \frac{\eta^2_{P1-CDS} f^2_{P1-CDS}}{\eta^2_{CDS}} \tag{8}$$

The rest of the intrinsic noise was contributed by P2.

**Measuring the expression patterns of *Kr* and *kni*.** We measured the expression patterns of *Kr* and *kni* using pre-gastrulation nc14 embryos. For each embryo, we plotted the nascent mRNA signal ($r$, in units of the number of molecules) against nuclear position ($x$) for all nuclei and binned individual data points by $x$. Within position ranges 0.3–0.8 EL (for *Kr*) or 0.4–1.0 EL (for *kni*), we used a least-squares algorithm (the "nlinfit" function in MATLAB) to fit the binned data to a product of two logistic functions:

$$r = r_{max} \frac{e^{(x-x_{0A})/d_A}}{(e^{(x-x_{0A})/d_A} + 1)} \frac{e^{-(x-x_{0P})/d_P}}{(e^{-(x-x_{0P})/d_P} + 1)} + r_0 \tag{9}$$

where $x_{0A}$ and $x_{0P}$ denote the anterior and posterior boundary positions of the expression domain, respectively.

**Measuring the gene regulation function.** To analyze the regulation of a promoter by Bcd, we plotted, for each embryo, the nascent mRNA signal versus Bcd concentration for each nucleus within the position range of 0.25–0.75 EL. To extract the GRF, we binned individual data points by Bcd concentration and fitted them to a Hill function using a least-squares algorithm (the "nlinfit" function in MATLAB):

$$y = a \frac{[Bcd]^h}{[Bcd]^h + C_0^h} + d \tag{10}$$

where $h$ is the Hill coefficient, $C_0$ is the concentration threshold for promoter activation, $a$ indicates the maximal level of Bcd-dependent activity, and $d$ denotes the basal activity.

**Measuring Bcd binding.** To quantify Bcd binding at a specific *hb* promoter, we used active P1 and P2 transcription sites to locate individual (active) promoter loci. Near each locus, a locus-integration region ($xy$ distance ≤3 pixel and $z$ distance = 0 from the locus) and an out-of-locus region ($xy$ distance ≤ 6 pixels and ≥ 3 pixels, $z$

distance = 0) were defined, which covered the nuclear volumes of $V_1$ and $V_o$, respectively. The nuclear IF signal (in units of the number of Bcd molecules) within these two regions was integrated and denoted as $I_l$ and $I_o$, respectively. The enriched Bcd signal was defined as the difference between $I_l$ and $I_o$ in consideration of the volume difference between the two regions, i.e., $I_{enrich} = I_l - I_o \cdot V_l / V_o$. Data from multiple embryos in the same nuclear cycle were pooled to increase the sample size. Data in the nuclear position range of 0.2–0.35EL were averaged to estimate the mean Bcd binding level of the anterior expression domain. To plot the Bcd binding curve, we binned the single-locus data by nuclear position and related mean Bcd enrichment with mean nuclear position or Bcd concentration.

**Mathematical modeling of transcriptional kinetics.** Stochastic modeling and inference of transcriptional kinetics are described in detail in Supplementary Note 6.

**Reporting summary.** Further information on research design is available in the Nature Research Reporting Summary linked to this article.

## Data availability
The raw image data reported in this paper are publicly accessible at a private server (http://gofile.me/4yuzx/wKna2V9pK). Source data are provided with this paper.

## Code availability
Custom scripts for data analysis and mathematical modeling were written in MATLAB 2018a (MathWorks) and are available in Github (https://github.com/Xulab-SJTU/Quantify-the-transcriptional-regulation) and in Zenodo (https://doi.org/10.5281/zenodo.6445280)[71].

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

## Acknowledgements

We thank Stephen Small, Pinar Onal, Jun Ma, Michael Perry, and Alistair Boettiger for the generous gift of fly lines. We thank Ido Golding, Anna Sokac, and Jun Ma for insightful discussion and valuable comments on the manuscript. This work was supported by the National Key R&D Program of China (grant no. 2018YFC0310803, 2021YFA0910702), the National Natural Science Foundation of China (grant no. 11774225, 41921006), the Natural Science Foundation of Shanghai (grant no. 18ZR1419800), and the Burroughs Wellcome Fund Career Award at the Scientific Interface (grant no. 1013907) to H.X. We gratefully acknowledge the imaging and computing resources provided by the Instrumental Analysis Center and the Student Innovation Center at Shanghai Jiao Tong University.

## Author contributions

Conceptualization by J.W. and H.X.; Methodology by J.W., H.L., S.Z., and H.X.; Software by S.Z. and H.X.; Formal Analysis by J.W. and S.Z.; Investigation by J.W., S.Z., and H.X.;

Writing – Original Draft by J.W. and H.X.; Writing – Revised Draft by J.W. and H.X.; Funding acquisition by H.X.; Resources by H.X.; and Supervision by H.X.

## Competing interests

The authors declare no competing interests.
