## [Peer Review File · Nature Communications]

Differential regulation of alternative promoters emerges from unified kinetics of enhancer-promoter interactionReviewers' Comments:

Reviewer #1:

Remarks to the Author:

The manuscript by Wang et al uses single molecule FISH (smFISH) to quantify the expression of hunchback (hb) that driven by the P1 and P2 promoters during the nuclear cycle 11 to 13 in *Drosophila* embryos. Unlike previous findings that the activator Bicoid (Bcd) only activates P2, the authors showed that both P1 and P2 promoters activate transcription in early embryos. Using a theoretical model and quantitative analysis, the authors analyzed how the two hb promoters interact differently with the two hb enhancers to regulate gene expression. P1 requires cooperative binding of Bcd and synergistic action of two enhancers, while P2 relies primarily on Bcd binding to one enhancer. The authors ask an interesting question of how multiple promoters can interact with multiple enhancers to regulate gene expression.

However, some of their claims are not necessarily supported by the data, because of the limitations in experimental design (described in detail below). Also, the contribution of P1 promoter is about 5-10% of the entire hb expression, and it was shown previously that P1-mediated transcription of hb does not affect embryonic patterning. In that case, it is unclear if the mechanism of which hb enhancers interact with the two promoters can be generalizable for other gene regulation. However, I agree that the quantitative framework the authors used to analyze the differential role of P1 and P2 in hb transcription is useful for other studies of enhancer-promoter interactions. I believe that this work needs a revision to be accepted in *Nature Communications*.

1. The authors claim that not all P2-driven transcriptional activity is visualized by the P2-3'UTR probes, because of the probe location. In that case, P2-3'UTR probe does not seem to be an ideal probe to be used to compare P1- and P2-driven transcriptional activities. Yet, the authors use P1-5'UTR and P2-3'UTR probes to examine differential role of P1 and P2 promoters. I am not sure if this is a fair comparison, given all unaccounted P2 transcripts.

Later, the authors concluded that majority of the "only CDS" transcripts are the P2-driven transcripts, because the hb expression boundary of the "only CDS" transcripts match more closely to the P2-driven transcripts than the P1-driven ones. In the supplementary Figure 2d, the boundary (x_0) of the "only CDS" is higher than P2 (except for Cycle 12). I am not convinced if this data alone can support the view that the unaccounted transcripts are all from the P2.

2. In Figure 4, the authors used yellow transcripts as a proxy for P2 – again, based on the assumption that most of the CDS transcription is driven by P2. Did the authors obtain similar results when they analyzed the difference in transcriptional activity between the P1- and P2-specific probes? Similarly, I think the analysis in Figure 5 using the P1 and P2 specific probes should also be done with the CDS probe. Since the authors argue that >90% of CDS activity corresponds to P2, it'll be helpful to use CDS as a proxy for P2 to further support their claim that the state 2 contributes less to P2 and it mainly contributes to P1.

3. As the authors said in the manuscript, the r_{max} of P2-3'UTR is low, because the probe targets the 3'UTR of hb. In that case, Figure 1e (nascent mRNA plot) can be somewhat misleading, because it gives the impression that the transcriptional activity driven by the P2 promoter is comparably low as the P1-driven transcription. And it is also inconsistent with high CDS transcriptional activity.

4. In Figure 1g and the supplementary Figure 2d, the boundary (x_0) was obtained by fitting a logistic function to the nascent mRNA plot. A representative fit to the logistic function would be needed to determine the goodness of the fit. In the Figure 1e for Cycle 13, P2-3UTR and the P1-5UTR curve look pretty much the same. However, there is a big difference in x_0 value shown in Figure 1g. This makes me doubt if the obtained boundary value is accurate.

5. In Figure 3f, the authors claim that more Bcd enrichment occurs in P1 compared to P2. However, statistical significance is shown only for the cycle 12, and not cycle 11 and 13.

6. In Figure 4g and h, the authors claim that for P2, the enhancers compete with each other, while the enhancers work synergistically for P1. This is based on the change in hb boundary. However, as the authors mentioned in the manuscript, transcriptional activity from P1 barely changed upon distal enhancer removal. Similarly, the P2-driven transcription shows significant posterior shift only for cycle 12. This data doesn't necessarily support the authors claim in enhancer competition or cooperativity.

7. The authors claim that hb transcription through P1 is not negligible and it affects the shape and fluctuation of the hb pattern. Based on their results, P1 contributes to about 5-10% of the hb amplitude, and about 1-cell width of the expression domain. It will be nice if the authors can further characterize or mention previous results that show how such subtle change can still affect subsequent patterning activity in embryonic development. It was shown by Ling et al (2019) that transcription from P1 is not required for hb's functions in embryonic patterning.

Reviewer #2:

Remarks to the Author:

The manuscript by Wang et al. investigates the contribution of alternative promoters and multiple enhancers to transcriptional regulation. For this, they employ the regulation of hb expression in the early *D. melanogaster* embryos as a paradigm.

Using smFISH and quantitative analysis, the authors characterize in depth the contribution of P1 and P2 promoters to hb endogenous expression in terms of timing, levels and precision of the border. First, the authors demonstrate that P1-driven expression is non-negligible and accounts for ~13% of total hb expression (and ~20% of hb noise) during early *Drosophila* development. Next, the authors dissect how this P1 promoter responds to Bcd concentration and conclude that P1 activation requires cooperative binding of more Bcd molecules than P2. Using published BAC lines, they analyze how the primary and shadow hb enhancers affect P1 expression. The manuscript ends by modeling P1/P2 transcription from smFISH data and estimates kinetic parameters of promoter switching.

While I find the data robust and most of the analysis well performed (see below), I am concerned about the functional relevance of this newly characterized transcription from P1. The manuscript would be much more exciting if we knew the function of the discovered P1-driven 13% contribution to hb expression, potentially by employing CRISPR mutations.

Specific points:

1-Most of the work presented aims at comparing transcriptional output from two alternative promoters. Yet this is solely examined by a single approach, promoter-specific smFISH. It would be important to confirm this expression bias by an alternative method (published CAGE data, 5'RACE on staged embryos or live imaging with the MS2/MCP technique).

2-The conclusion regarding cooperative binding of bcd as a regulator of P1 expression is primarily based on the estimation of the Hill coefficient from fixed tissues. It would be important to back up this conclusion with manipulations in cis (bcd binding site mutants) or in trans (optogenetic tool to fine tune bcd levels exist : Huang et al 2017).

In particular, interpretation of Bcd protein quantification close to P1/P2 loci from fixed tissues should be nuanced. First because, as discussed in a recent paper (Fernandes et al., biorxiv 2021) Bcd concentration does not reflect functional bcd gradient. Second, because Bcd was shown to form dynamic clusters with transient binding kinetics (RT in the order of seconds), which depend on the presence of proximal ZDL clusters (Mir et al., 2017, 2018). Thus quantification of bcd levels at a specific location at a specific time may not capture the dynamic aspects.

Technical points :

- How is nascent transcription estimated when nuclei show 3 or 4 spots (sister chromatids). The methods section indicates how sister loci pairs were distinguished from unpaired loci (line 482), but does not explain if the quantification was restricted to nuclei in which paired sister chromatids are detected. Moreover, the choice for distance thresholds is not argued. The added complexity of early replication and potential presence of 3 or 4 spots should be mentioned in the main text (line 97).
- To compare signals from different probes, it is essential to verify that the efficiency of fluorescent detectors are comparable. It might be useful to switch fluorescent detectors between P1 and P2.

Modeling :

- The choice for a 3 state model is not very clearly explained (line 251)
In addition, why a 3 state model with 2 ON/1 OFF is envisaged rather than 2 OFF and 1 ON state (as shown in the context of *Drosophila* embryos in Pimmett, Dejean et al., 2021)?
Indeed, while it's easy to envisage various biochemical state for inactive states, what would distinguish the two proposed active states ?
This point could be discussed
- Regarding promoter regulation by more than 2 states, other references could be considered Innocentini et al., 2013 ; Tantale et al., 2016; Pimmett et al., 2021.

Minor points :

- The last sentence of the abstract line is more strongly phrased than the conclusions from the data allow. Since the entire manuscript solely uses fixed imaging that doesn't capture the dynamic aspects of gene regulation, line 27 should be reformulated
- Line 125 : mention in the main text that the Fano factor is calculated for nascent mRNAs and not on cytoplasmic mRNA.
- Sup1 b : there is a '19' number in the middle of the graph
- Name with a panel the FISH image + quantification in the bottom part of Figure 1b.
Explain more in the figure legend how the data were 'binned'.
- Can the authors explain why the residency time for P1 transcript is 3 times longer than that of P2 transcript. The supp method section (page 4) indicates $T_r = 142$ for P1 and $T_r = 46$ sec for P2. Where do these estimate come from ?

DETAILED ANSWERS TO REVIEWER COMMENTS:

We thank both reviewers for their valuable and constructive comments. Below we provide a point-by-point response. The comments are shown in *blue*, and our responses are shown in black. For clarity, figure names are in **red**, section names are in **green**, and references are in **dark blue**.

** Due to the COVID-19 pandemic, our international chemical orders (e.g., FISH probes) suffered from long-term delays. To finish the revision on time, we used HCR-FISH (probes were previously ordered for other projects) as a replacement of smFISH to identify active yellow and lacZ loci in some control experiments (see Methods and Supplementary Note 2). These signals were only used for qualitative purposes. All quantitative analyses were based on smFISH signals.*

Reviewer #1

The manuscript by Wang et al uses single molecule FISH (smFISH) to quantify the expression of hunchback (hb) that driven by the P1 and P2 promoters during the nuclear cycle 11 to 13 in Drosophila embryos. Unlike previous findings that the activator Bicoid (Bcd) only activates P2, the authors showed that both P1 and P2 promoters activate transcription in early embryos. Using a theoretical model and quantitative analysis, the authors analyzed how the two hb promoters interact differently with the two hb enhancers to regulate gene expression. P1 requires cooperative binding of Bcd and synergistic action of two enhancers, while P2 relies primarily on Bcd binding to one enhancer. The authors ask an interesting question of how multiple promoters can interact with multiple enhancers to regulate gene expression.

However, some of their claims are not necessarily supported by the data, because of the limitations in experimental design (described in detail below). Also, the contribution of P1 promoter is about 5-10% of the entire hb expression, and it was shown previously that P1-mediated transcription of hb does not affect embryonic patterning. In that case, it is unclear if the mechanism of which hb enhancers interact with the two promoters can be generalizable for other gene regulation. However, I agree that the quantitative framework the authors used to analyze the differential role of P1 and P2 in hb transcription is useful for other studies of enhancer-promoter interactions. I believe that this work needs a revision to be accepted in Nature Communications.

Response:

We thank the reviewer for pointing out the limitations in our experimental design. Following the detailed suggestions, we have done additional experiments and data analysis to improve the support of our conclusions (see below for details). Moreover, we have used a CRISPR-based P1 deletion fly line to show that early P1 transcription does affect the patterning of other segmentation genes (see the response to Reviewer #1's last comment). Thus, the modest P1 expression in early development is biologically meaningful. Such weak promoter activity may exist in many eukaryotic genes containing multiple promoters and enhancers. A complete understanding of gene behavior not only includes knowledge about the strongly expressed promoters but also requires an accurate picture of the weakly expressed ones. Specifically, why are these promoters expressed differently?

What are their biological functions? Our study not only investigated these questions for a specific gene, but also provided a general mechanistic framework for understanding the regulation dynamics and functions of other multi-promoter genes.

1. The authors claim that not all P2-driven transcriptional activity is visualized by the P2-3'UTR probes, because of the probe location. In that case, P2-3'UTR probe does not seem to be an ideal probe to be used to compare P1- and P2-driven transcriptional activities. Yet, the authors use P1-5'UTR and P2-3'UTR probes to examine differential role of P1 and P2 promoters. I am not sure if this is a fair comparison, given all unaccounted P2 transcripts.

Later, the authors concluded that majority of the “only CDS” transcripts are the P2-driven transcripts, because the hb expression boundary of the “only CDS” transcripts match more closely to the P2-driven transcripts than the P1-driven ones. In the supplementary Figure 2d, the boundary (xo) of the “only CDS” is higher than P2 (except for Cycle 12). I am not convinced if this data alone can support the view that the unaccounted transcripts are all from the P2.

Response:

The reviewer's concern is reasonable since the probe location heavily affects the FISH signal of nascent transcripts. In general, probes targeting the 5' region (except the intron) of the gene can label more nascent transcripts and give a brighter signal than that targeting the 3' region. For this reason, most FISH studies used 5'UTR or CDS probes. However, labeling the 3' region of the transcript has also been tried, whose signal was, on average, proportional to that of 5' probes ((Zoller et al., 2018); see a theoretical proof in Supplementary Note 6.5). Thus, with a proper scaling factor (or ratio parameter), the FISH signal of 3' probes may be used for quantitative analysis (i.e., compare with the signal of 5' probes).

In this paper, we tried to distinguish P1 and P2 activities using transcript-specific FISH probes. For the P1-driven transcript, we used probes targeting its unique 5'UTR and intron. In contrast, P2-3'UTR probes are the only choice for specific labeling of the P2-driven transcript, since all other regions of the transcript share sequence with its P1-driven isoform. Due to the difference in target locations (5' vs. 3'), signals of P1 and P2 probes were not directly comparable. In the original manuscript, we tried to solve this issue by converting signals of both probe sets to the equivalent CDS signals from the corresponding promoters. This was done by decomposing the nascent CDS signal of P1/P2 double active loci as a linear combination of P1-5'UTR and P2-3'UTR signals in the embryo labeled with the three probe sets. This method enabled a quantitative comparison between the activities of the two promoters in a common unit. However, since the “only CDS” transcripts were excluded from the analysis, the decomposition may be incomplete.

In the revised manuscript, to verify the general idea of the above method, we have directly examined the relationship between different probe signals of the same promoter. We use CRISPR mutant fly lines with P1 and P2 deletions ($\Delta P1C$ and $\Delta P2C$ (Ling et al., 2019)) obtained from Dr. Stephen Small (New York University) and Dr. Pinar Onal (Northwestern University). By labeling

these mutant embryos with P1-5'UTR, P2-3'UTR, and CDS probes, we confirm that P1-5'UTR and P2-3'UTR probes are promoter-specific (**Supplementary Fig. 4a**). For either mutant, the average signals of transcript-specific probes are proportional to that of CDS probes in the anterior expression domain (**Supplementary Fig. 4b**). Moreover, we find that CDS and P1-5'UTR probes label more nascent mRNA foci than their 3' counterparts (P2-3'UTR and CDS) in $\Delta P1C$ and $\Delta P2C$ embryos, respectively (**Supplementary Fig. 4c**). This result shows that most of the “only CDS” nascent transcripts are P2-driven. In the revised manuscript, this experiment is presented in **Supplementary Note 3** and cited by the second section of the **Results** part (pages 8–9).

Following the above experiment, we have further improved the linear decomposition method for the wild-type embryo. By locally averaging the nascent signal over neighboring nuclei, we are able to include the “only CDS” transcripts into the analysis. With a linear decomposition of the locally averaged nascent CDS signal into P1-5'UTR and P2-3'UTR ones, we find the ratio parameter between the P1-5'UTR and CDS signals unchanged ($a_1 = 0.53 \pm 0.03$, **Fig. 2c**). Thus, the estimated P1 contribution to early *hb* transcription remains similar as before (**Fig. 2e–f**). In contrast, the ratio parameter between the CDS and P2-3'UTR signals is now $a_2 = 3.72 \pm 0.90$ (**Fig. 2c**), much larger than the previous value ($a_2 = 2.74 \pm 0.16$). The increase in a_2 enables an almost complete decomposition of the nascent CDS signal into P1-5'UTR and P2-3'UTR signals in the anterior expression domain (0.2–0.4 EL, **Supplementary Fig. 5c**). It confirms that most of the “only CDS” signal comes from P2. Note that the above a_1 and a_2 values do not explain all CDS signals in terminal regions (0–0.2 and 0.8–1 EL; **Supplementary Fig. 5c**), suggesting different a_1 and a_2 for these regions. A side effect of the increased a_2 is that the estimated post-elongation residence time (T_R) of P2-driven transcripts decreases from 46 s to 22 s. The new T_{R-P2} agrees better with the previous study (<35 s (Zoller et al., 2018)) and does not significantly affect most other findings of the paper. In the revised manuscript, we update the second section of the **Results** part (page 9) with the above analysis. Some technical details are presented in section “**Estimating promoter contributions to *hb* activity**” of the **Methods** part (pages 33–34).

In addition to the mean expression level, the statistical properties of the 5' and 3' probe signals for the same promoter are also intrinsically related. The distributions of the two signals correspond to the same transcription kinetics, which may be extracted from either distribution using probe-specific contribution functions (see **Supplementary Note 6**). In the revised manuscript, we have verified the above argument by analyzing the distributions of nascent CDS signal and comparing the extracted kinetics with that from nascent P2-3'UTR signal (see below the response to the next comment).

2. *In Figure 4, the authors used yellow transcripts as a proxy for P2 – again, based on the assumption that most of the CDS transcription is driven by P2. Did the authors obtain similar results when they analyzed the difference in transcriptional activity between the P1- and P2-specific probes? Similarly, I think the analysis in Figure 5 using the P1 and P2 specific probes should also be done with the CDS probe. Since the authors argue that >90% of CDS activity corresponds to P2, it'll be helpful to use CDS as a proxy for P2 to further support their claim that the state 2 contributes less to P2 and it mainly contributes to P1.*

Response:

Following the reviewer's comment on **Fig. 4**, we have done a complementary experiment, in which enhancer deletion embryos (and their controls) were labeled with P1-5'UTR, P2-3'UTR, and *yellow* probes. P1-5'UTR and P2-3'UTR signals corresponding to the transgene were identified based on their colocalization with *yellow* foci. Results of this experiment are presented in **Supplementary Fig. 8** of the revised manuscript (cited by the fourth section of the **Results** part, page 13). In general, the results are similar to those in the original manuscript. I.e., (1) removing either enhancer significantly reduces the active percentage and expression amplitude of P1 and P2 (**Supplementary Fig. 8c–f**, also see the response to comment #6 for our improvements to **Fig. 4**). (2) Proximal and distal enhancer removals shift the P2 expression boundary to opposite directions, while both deletions cause an anterior shift of the P1 expression boundary (**Supplementary Fig. 8g**). A slight difference between **Fig. 4g** and **Supplementary Fig. 8g** is on the magnitude of the boundary shift. This may be because the reporter gene is too long (~10 kbp) and does not have enough time to reach steady-state transcription in a nuclear cycle. Specifically, when its P1-5'UTR signal reaches a high value, its P2-3'UTR signal may just emerge. When its P2-3'UTR signal reaches a high value, its P1-5'UTR signal may have diminished due to transcription shut down at the end of the nuclear cycle. In such a case, labeling the intron and CDS (*yellow*) regions may be a better choice, as they are relatively close to each other.

As for the analysis in **Fig. 5**, we have applied it to the CDS signal. The distribution of nascent CDS signal fits a three-state model as well (**Supplementary Fig. 10c**). The extracted kinetic rates and their dependence on Bcd are similar to that from the P2-3'UTR signal (**Supplementary Fig. 10d**). Thus, CDS may be used as a proxy for P2. Quantitatively, k_{01} is much larger than k_{12} , supporting our conclusion that P2 activation mainly relies on state 1. k_{12} estimated from the CDS signal is bigger than that from the P2-3'UTR signal (~0.10 min⁻¹ vs. ~0.06 min⁻¹), revealing an influence from P1. In the revised manuscript, these results are cited by the fifth section of the **Results** part (page 16).

*3. As the authors said in the manuscript, the r_{max} of P2-3'UTR is low, because the probe targets the 3'UTR of *hb*. In that case, Figure 1e (nascent mRNA plot) can be somewhat misleading, because it gives the impression that the transcriptional activity driven by the P2 promoter is comparably low as the P1-driven transcription. And it is also inconsistent with high CDS transcriptional activity.*

Response:

We apologize for not explaining **Fig. 1e–f** well in the original manuscript. As we mentioned in the response to the reviewer's first comment, 5' probes are expected to give a brighter signal than their 3' counterparts. Thus, it is not surprising that r_{max} of P2-3'UTR is much lower than that of CDS. For this reason, the magnitudes of P1-5'UTR and P2-3'UTR profiles should not be directly compared without proper scaling. Following this rule, we did not compare r_{max} between P1-5'UTR and P2-3'UTR in the first section of the **Results** part. Instead, we compared r_{max} between P1-5'UTR and

CDS (as a proxy for P2) to roughly show that P2 is more active than P1 in early development. Then, with proper scaling parameters for P1-5'UTR and P2-3'UTR, we quantitatively compared P1 and P2 activities in the second section of the **Results** part (see the response to comment #1).

In the revised manuscript, to avoid misunderstanding of **Fig. 1e–f**, we have modified the relevant description in the first section of the **Results** part (pages 7–8) as follows:

As expected, r_{max} of P1-specific signals was much smaller than that of the CDS signal (Fig. 1f), indicating that P1 contributes less than P2 to early hb expression. r_{max} of the P2-specific signal also remained low (Fig. 1f) due to the probes' target location at the 3' end of hb-RA. r_{max} of the P1- and P2-specific signals should not be directly compared, since their probes target different ends of nascent transcripts (5' vs. 3'; see next section for a comparison between P1- and P2-specific signal levels).

4. In Figure 1g and the supplementary Figure 2d, the boundary (x_0) was obtained by fitting a logistic function to the nascent mRNA plot. A representative fit to the logistic function would be needed to determine the goodness of the fit. In the Figure 1e for Cycle 13, P2-3UTR and the P1-5UTR curve look pretty much the same. However, there is a big difference in x_0 value shown in Figure 1g. This makes me doubt if the obtained boundary value is accurate.

Response:

Following the reviewer's comment, we have added a supplementary figure panel to show a representative fit to the logistic function for each probe signal (**Supplementary Fig. 3b**). The estimated R-square, a measure of the goodness of fit, is >0.93 for all embryos, indicating good fits. As for **Fig. 1e**, it may be difficult to directly compare the boundary positions (x_0) of different curves by eye, because these curves differ significantly in the amplitude (the maximal transcription level, r_{max}). To facilitate the comparison, we have rescaled the amplitude of every curve to one and plotted the transition region (0.3–0.6 EL) of the rescaled curves as insets for **Fig. 1e**. We can easily observe the difference (in x_0) between P2-3UTR and P1-5UTR in all cycles, including cycle 13. This result is consistent with **Fig. 1g**.

5. In Figure 3f, the authors claim that more Bcd enrichment occurs in P1 compared to P2. However, statistical significance is shown only for the cycle 12, and not cycle 11 and 13.

Response:

We apologize for the confusion. The statistical significance of a result is affected by sample size. In the original **Fig. 3f**, the average Bcd enrichment per active promoter loci was computed from ~5 embryos for each cycle. However, there are much fewer nuclei (and fewer gene loci) in an nc11 embryo than in an nc12-13 embryo. To ensure a fair comparison, we have measured three more embryos in nc11. Without affecting the mean values of Bcd enrichment, the increased sample size

induces statistical significance ($p < 0.05$) to nc11 data (see the updated **Fig. 3f**). In contrast, the lack of statistical significance for nc13 data is not due to the small sample size but rather caused by a decrease in the P1-specific enrichment plateau (from ~8–10 Bcd molecules to ~6 Bcd molecules) at later developmental stage (**Supplementary Fig. 7**). Such a change is consistent with the observation that the Hill coefficient of k_{12} drops from ≥ 8 to ~6 in nc13 (**Fig. 5d**). In the original manuscript, we briefly discussed the change in Bcd enrichment in nc13 at the end of the third section of the **Results** part (unchanged in the revised manuscript, now on pages 12–13).

6. In Figure 4g and h, the authors claim that for P2, the enhancers compete with each other, while the enhancers work synergistically for P1. This is based on the change in hb boundary. However, as the authors mentioned in the manuscript, transcriptional activity from P1 barely changed upon distal enhancer removal. Similarly, the P2-driven transcription shows significant posterior shift only for cycle 12. This data doesn't necessarily support the authors claim in enhancer competition or cooperativity.

Response:

Following the reviewer's concern about **Fig. 4**, we have double-checked the raw data and the analysis procedure. We find that the unexpected behaviors of P1 and P2 are due to the following issues in data analysis:

- (1) Our previous quantification of the single-locus transcription level may be affected by inaccurate estimation of nuclear gene copy number. Following Reviewer #2's comment (#3), we notice that accurate counting of nuclear gene copy number is critical for single-locus analysis. Specifically, a nucleus may contain two or four copies of a gene (for homozygous alleles) during the mitotic interface. Yet, we are not always certain about the exact gene copy number by analyzing the FISH image. Nuclei with active sister loci pairs (typically containing ≥ 3 active loci) certainly have four gene copies. However, the gene copy number for nuclei without active sister loci pairs (typically containing < 3 active loci) is unknown. In the original manuscript, we coarsely assumed that these nuclei each had two gene copies. However, such inference may not be 100% accurate and may affect the quantification of promoter expression level. To avoid the possible inaccuracy in gene copy inference, we have summed over the nascent signal in each nucleus as an alternative measure of the average promoter activity in the revised manuscript (applied to all mean-level analysis in the first four sections of the **Results** part).
- (2) We find that two of the control strain embryos in nc12 were actually in the early stage of the mitotic interface (based on the nuclear shape). The expression levels of these embryos have not reached a steady state. It is against the rule of embryo selection used in this paper (we choose embryos in the mid-to-late mitotic interphase. See the **Methods** part, page 30). We now exclude these embryos from our data set. We have also double-checked all embryos used in this paper to ensure they are all in the right stage.

With the above improvements, we have redone the analysis and replotted **Fig. 4** for the fourth

section of the **Results** part (pages 13–14). In the new version of **Fig. 4c–f**, P1-driven transcription per nucleus shows a clear decrease upon distal enhancer removal for nc11-12. Meanwhile, P2-driven transcription per nucleus shows a more significant posterior shift upon distal enhancer removal for all cycles in the new version of **Fig. 4g**. These results support our conclusion about different enhancer action modes for P1 and P2.

7. The authors claim that hb transcription through P1 is not negligible and it affects the shape and fluctuation of the hb pattern. Based on their results, P1 contributes to about 5-10% of the hb amplitude, and about 1-cell width of the expression domain. It will be nice if the authors can further characterize or mention previous results that show how such subtle change can still affect subsequent patterning activity in embryonic development. It was shown by Ling et al (2019) that transcription from P1 is not required for hb's functions in embryonic patterning.

Response:

We apologize for not investigating the developmental consequence of early P1 activity in the original manuscript. In the previous literature, P1 was believed to be inactive and thus irrelevant to anterior-posterior patterning during early development (Margolis et al., 1995; Wu et al., 2001). Specifically, by constructing a CRISPR/Cas9-based P1 mutant ($\Delta P1C$), Ling et al. showed that deleting P1 did not disrupt the larval cuticle (Ling et al., 2019). They concluded that transcription from P1 is not required for *hb*'s functions in embryonic patterning. This conclusion is more or less reasonable since the majority of early *hb* expression is from P2.

However, cuticle morphology is only one (out of many) downstream aspect of embryonic patterning that involves many upstream inputs besides *hb*. To characterize the direct consequence of early P1 transcription, we have measured the expression profiles of *hb*-target genes *Krüppel* (*Kr*) and *knirps* (*kni*) in $\Delta P1C$ fly line (obtained from Dr. Stephen Small (New York University) and Dr. Pinar Onal (Northwestern University)) using smFISH. According to previous studies, the anterior borders of *Kr* and *kni* expression domains in nc14 are both set by the repression of Hb protein gradient (Hulskamp et al., 1990; Struhl et al., 1992). Consistent with this picture, we observe a significant anterior shift of the medial expression domain for *Kr* and *kni* in response to P1 deletion (**Fig. 2g, h**). The anterior *Kr* boundary is shifted for ~ 0.09 EL (from 0.45 ± 0.02 EL in WT to 0.36 ± 0.01 EL in $\Delta P1C$, $p < 0.01$), while the anterior *kni* boundary is shifted for ~ 0.06 EL (from 0.58 ± 0.01 EL in WT to 0.52 ± 0.02 EL in $\Delta P1C$, $p < 0.05$). These results show that early P1 transcription can quantitatively affect the subsequent patterning activity in embryonic development.

In the revised manuscript, we write a new paragraph to present the above results at the end of the second section of the **Results** part (page 10). It should be noted that these results are not against Ling et al. but instead focus on an earlier step of the patterning process. The difference between the two studies suggests that inputs from other genes may compensate for the missing P1 in the downstream patterning process. Filling the gap between the two studies will be an interesting topic for future work.

Reviewer #2

*The manuscript by Wang et al. investigates the contribution of alternative promoters and multiple enhancers to transcriptional regulation. For this, they employ the regulation of *hb* expression in the early *D. melanogaster* embryos as a paradigm.*

*Using smFISH and quantitative analysis, the authors characterize in depth the contribution of P1 and P2 promoters to *hb* endogenous expression in terms of timing, levels and precision of the border. First, the authors demonstrate that P1-driven expression is non-negligible and accounts for ~13% of total *hb* expression (and ~20% of *hb* noise) during early *Drosophila* development. Next, the authors dissect how this P1 promoter responds to Bcd concentration and conclude that P1 activation requires cooperative binding of more Bcd molecules than P2. Using published BAC lines, they analyze how the primary and shadow *hb* enhancers affect P1 expression. The manuscript ends by modeling P1/P2 transcription from smFISH data and estimates kinetic parameters of promoter switching.*

*While I find the data robust and most of the analysis well performed (see below), I am concerned about the functional relevance of this newly characterized transcription from P1. The manuscript would be much more exciting if we knew the function of the discovered P1-driven 13% contribution to *hb* expression, potentially by employing CRISPR mutations.*

Response:

We thank both reviewers for inquiring about the functional relevance of early P1 activity. On the last page, we have responded to this comment in detail. Here we restate the response as follows:

Traditionally, P1 was believed to be inactive in early development and thus irrelevant to *hb*'s functions in embryonic pattern formation (Margolis et al., 1995; Wu et al., 2001). For instance, Ling et al. employed a CRISPR/Cas9-based P1 mutant (Δ P1C) to show that deleting P1 did not cause detectable disruption of the larval cuticle (Ling et al., 2019). They concluded from this observation that P1 activity is not required for embryonic patterning. This conclusion is reasonable to some extent since early *hb* expression is mainly from P2.

However, cuticle morphology is only one (out of many) downstream aspect of embryonic patterning that involves many upstream inputs besides *hb*. To characterize the direct consequence of early P1 transcription, we have obtained Δ P1C fly line from Dr. Stephen Small (New York University) and Dr. Pinar Onal (Northwestern University) and have applied smFISH to measure the expression profiles of *hb*-target genes *Krüppel* (*Kr*) and *knirps* (*kni*) in early development. Based on the literature, the anterior borders of *Kr* and *kni* expression domains in nc14 are both set by the repression of Hb protein gradient (Hulskamp et al., 1990; Struhl et al., 1992). Consistent with this picture, we observe a significant anterior shift of the medial expression domain for *Kr* and *kni* in response to P1 deletion (Fig. 2g, h). The anterior *Kr* boundary is shifted for ~0.09 EL (from 0.45 ± 0.02 EL in WT to 0.36 ± 0.01 EL in Δ P1C, $p < 0.01$), while the anterior *kni* boundary is shifted for ~0.06 EL (from 0.58 ± 0.01 EL in WT to 0.52 ± 0.02 EL in Δ P1C, $p < 0.05$). These results reveal that early P1 transcription has a non-negligible effect on the patterning of other segmentation genes.

In the revised manuscript, we write a new paragraph to present the above results at the end of the second section of the **Results** part (page 10). It should be noted that the results from us and Ling et al. are not in contradiction. Instead, the two focused on different steps of the patterning process. The difference between the two studies suggests that, in the downstream patterning process, inputs from other genes may compensate for the missing P1. Further uncovering the mist between the two studies will be an interesting topic for future research.

Specific points:

1. Most of the work presented aims at comparing transcriptional output from two alternative promoters. Yet this is solely examined by a single approach, promoter-specific smFISH. It would be important to confirm this expression bias by an alternative method (published CAGE data, 5'RACE on staged embryos or live imaging with the MS2/MCP technique).

Response:

We agree that the comparison between the two promoters should be confirmed using alternative methods. Following the reviewer's suggestion, we have backed up our comparison with multiple approaches in the revised manuscript:

- (1) The published CAGE and 5'RACE data from the Berkeley Drosophila Transcription Network Project (<http://gbrowse.modencode.org/>) show that *hb* mRNAs in early embryos are transcribed from two starting sites corresponding to P1 and P2, respectively (**Supplementary Fig. 1a, b**). This agrees with our finding that both promoters are active in early development.
- (2) According to previous studies, P1 and P2 transcripts terminate at different sites in the genome (<http://flybase.org>) (Driever and Nusslein-Volhard, 1989; Tautz et al., 1987). To validate this knowledge, we have done 3'RACE on stage 4-5 embryos (**Supplementary Fig. 1c**). Using a specifically designed protocol to avoid hybridization between P1 and P2 transcripts in PCR (protocol described in **Supplementary Note 1**), we confirm the existence of P1-driven transcripts in the early embryo. In addition, the result directly shows that P2 transcripts terminate at a more distant site than P1 transcripts. It thus validates the usage of P2-3'UTR probes.
- (3) The published RNA-seq and PolII ChIP-seq data for the early *Drosophila* embryo (<https://gander.wustl.edu/>, data were from the Gene Expression Omnibus database at NCBI (accession number: GSE41700) and the ENCODE data portal at UCSC, respectively) both exhibit peaks at P1 and P2 (**Supplementary Fig. 1d, e**). The peak at P1 is much lower than that at P2, suggesting that P1 is less active than P2 in the early embryo. This result is consistent with our finding that P2 contributes more than P1 in early *hb* transcription.

In the revised manuscript, we present the above results in **Supplementary Fig. 1** and cite them in the first section of the **Results** part (page 6). However, these sequencing-based methods typically

require RNA or DNA extraction from multiple embryos. Thus, they lack the sensitivity and spatial resolution for understanding the mechanism of transcriptional regulation at the single-cell and single-molecule level. In contrast, single-cell fluorescent imaging-based methods are suitable for this purpose. In particular, live imaging of MS2/MCP labeled mRNA directly detects transcriptional dynamics in the fly embryo. In recent years, this method was widely applied to uncover detailed dynamic features of many segmentation genes (Bothma et al., 2015; Garcia et al., 2013; Lim et al., 2018; Lucas et al., 2013; Pimmett et al., 2021). For example, MS2/MCP imaging of *hb* P2 activity with enhancer deletion revealed enhancer competition for P2 activation (Bothma et al., 2015), consistent with our results. However, the MS2/MCP method requires genetic modification of the endogenous gene, which may affect gene activity. In particular, the proximal enhancer of *hb* is embedded in the intron region downstream of P1. Therefore, inserting MS2 sequences into the P1-mRNA sequence may affect *hb* regulation. For this reason, we decided to image fixed wild-type embryos using smFISH, which can quantify the activity of endogenous P1 and P2 without genetic modification.

2. The conclusion regarding cooperative binding of bcd as a regulator of P1 expression is primarily based on the estimation of the Hill coefficient from fixed tissues. It would be important to back up this conclusion with manipulations in cis (bcd binding site mutants) or in trans (optogenetic tool to fine tune bcd levels exist : Huang et al 2017).

In particular, interpretation of Bcd protein quantification close to P1/P2 loci from fixed tissues should be nuanced. First because, as discussed in a recent paper (Fernandes et al., biorxiv 2021) Bcd concentration does not reflect functional bcd gradient. Second, because Bcd was shown to form dynamic clusters with transient binding kinetics (RT in the order of seconds), which depend on the presence of proximal ZDL clusters (Mir et al., 2017, 2018). Thus quantification of bcd levels at a specific location at a specific time may not capture the dynamic aspects.

Response:

The reviewer is correct that the conclusion regarding cooperative Bcd binding as a regulator of P1 expression needs validation using multiple approaches. In the original manuscript, besides estimating the Hill coefficients of P1 (and P2) from wild-type embryos, we also quantified Bcd binding by directly measuring local enrichment of the Bcd signal in the vicinity of P1- and P2-active *hb* loci. The promoter-specific binding curves exhibited discrete plateaus at ~4–5 and ~8–10 Bcd molecules, each implying a representative Bcd binding state for promoter activation. Since the numbers of bound Bcd molecules for these states are close to the Hill coefficients of P1 and P2, they directly support the picture of cooperative Bcd binding. In addition to the above studies of the wild-type embryo, we can manipulate the system *in cis* and *in trans* to further confirm this picture as follows:

(1) In the original manuscript, we mutated the *cis*-regulatory sequences to examine the roles of the proximal and distal enhancers in P1 (and P2) activation (see the fourth section of the **Results** part). Previous studies showed that each of these enhancers contains multiple Bicoid binding

sites and can independently drive P2 expression (Bothma et al., 2015; Perry et al., 2011). Specifically, it is believed that cooperative binding of Bcd at six canonical binding sites on the proximal enhancer is responsible for the Hill coefficient ($h \approx 5-6$) of P2. As a validation, mutating these sites in proximal-enhancer-only reporters was shown to reduce the Hill coefficient of P2 (Park et al., 2019). In contrast, the Hill coefficient of P1 is bigger than six ($h = 7.1 \pm 0.4$), which cannot be explained by Bcd binding at a single (proximal or distal) enhancer. In the revised manuscript, to explore the origin of this high Hill coefficient, we have measured the change of the regulatory relation between Bcd and P1 activity in response to enhancer removal (Supplementary Fig. 9e). By deleting the distal enhancer that contains multiple Bcd binding sites, the Hill coefficient of P1 drops significantly from ~ 6.6 to ~ 4.4 ($p < 0.05$), a value close to the Hill coefficient of P2. This result suggests that the high Hill coefficient of P1 is due to the cooperative binding of Bcd at both enhancers. It agrees with our conclusions in the fourth and fifth sections of the Results part. In contrast, the Hill coefficient of P2 does not change significantly with enhancer removal, consistent with the canonical picture of independent enhancer action for P2 regulation. These results are now presented in a new paragraph at the end of the fourth section of the Results part (page 15).

(2) In the original manuscript, we examined the role of Bcd in P1 (and P2) activation using embryos with $1\times$ maternally functional *bcd* copy (see the first paragraph of the third section of the Results part). Such manipulation of Bcd dosage (*in trans*) caused a significant shift of P1 (and P2) expression domain towards the anterior pole (Fig. 3a) by ~ 0.07 EL ($p < 0.01$, Fig. 3b), indicating that Bcd is an activator for both P1 and P2. In the revised manuscript, to further validate that Bcd activation of P1 is through cooperative binding, we have analyzed P1 (and P2) regulation in this fly line in more detail. Specifically, the estimated Hill coefficients of P1 and P2 in this fly line are similar to that in the wild-type embryo (Supplementary Fig. 6c). Namely, reducing the Bcd dosage does not affect the shape of GRF. This result is consistent with the cooperative-binding picture. The result is now presented in the third section of the Results part (page 12).

As for the concern about the nonfunctional component in Bcd gradient, we thank the reviewer for pointing out the recent paper from Fernandes et al. In this paper, the authors reported that the expression boundary of Bcd-driven genes shifted anteriorly by $\sim 10\%-11\%$ EL in response to reduction of maternal Bcd dosage from $2\times$ to $1\times$. Such a shift suggests an effective Bcd gradient with a decay length of $15.0\% \pm 1.4\%$ EL, significantly smaller than the value estimated from direct quantification of Bcd concentration in previous studies ($\sim 20\%$ EL (Gregor et al., 2007; Houchmandzadeh et al., 2002)). Thus, the authors proposed that only a sub-population of Bcd is functional.

However, it should be noted that most previous measurements of Bcd gradient were based on live imaging of Bcd-EGFP, which only gives fluorescence after a slow maturation of EGFP. Since Bcd molecules are synthesized at the anterior pole of the embryo, many anterior Bcd-EGFPs may be too young to be visible. As a result, the measured Bcd gradient may be milder than the actual one. In a previous paper by Liu et al. (Liu et al., 2013), this effect was estimated to cause a $\sim 15\%$ increase of the measured decay length. Thus, quantitative application of live Bcd data typically requires a correction of the EGFP maturation effect (Liu et al., 2013; Yang et al., 2020; Zhu et al., 2020).

Besides live imaging, Bcd gradient may be measured from fixed embryos using immunofluorescence. Without the EGFP maturation issue, this method typically reported a significantly smaller length constant (~16% EL (Cheung et al., 2011; Liu et al., 2013; Liu and Ma, 2015)). An exception is (Houchmandzadeh et al., 2002)). In this work, we used immunofluorescence to label Bcd in fixed wild-type embryos. The estimated decay length is $15.9\% \pm 0.5\%$ EL (Supplementary Fig. 6b), which agrees well with the effective decay length proposed by Fernandes et al. It suggests that our immunofluorescence data does reflect the functional Bcd gradient. The above result and discussion about Bcd quantification are now presented in Supplementary Note 4 and cited by the third section of the Results part (page 11).

Regarding the dynamic aspects of the local Bcd signal, we thank the reviewer for pointing out the pioneer works by Mir et al. (Mir et al., 2017; Mir et al., 2018). With live imaging, these works showed that Bcd and Zelda combinatorically bind DNA to form dynamic clusters, which transiently interact with Bcd-target genes to activate transcription. However, accurate quantification of Bcd binding dynamics using live imaging is still technically challenging to date due to the low signal level and slow maturation of the fluorescence protein. In this work, we measured local enrichment of Bcd at P1- and P2-active loci in fixed embryos, which corresponds to a snapshot of the Bcd cluster in the vicinity of *hb* loci at the time of fixation. Although our measurement did not directly capture the temporal information of Bcd binding, it is accurate enough for estimating the number of bound Bcd molecules. By computing the average level of Bcd binding in different Bcd concentration ranges, we recognized two typical Bcd binding states for *hb* activation. By relating these binding states with the statistics of promoter transcription, we reconstructed the dynamics of Bcd binding and *hb* promoter activation using a mathematical model. Such understanding of Bcd binding matches the focus of this study and may inspire future research of transcription factor binding. The above content is now presented in Supplementary Note 5 and cited by the third section of the Results part (page 12).

Technical points:

3. How is nascent transcription estimated when nuclei show 3 or 4 spots (sister chromatids).

The methods section indicates how sister loci pairs were distinguished from unpaired loci (line 482), but does not explain if the quantification was restricted to nuclei in which paired sister chromatids are detected. Moreover, the choice for distance thresholds is not argued. The added complexity of early replication and potential presence of 3 or 4 spots should be mentioned in the main text (line 97).

Response:

We thank the reviewer for pointing to the question of gene replication, which was not explained in detail in the original manuscript. Briefly speaking, nuclei with and without active sister loci pairs were both analyzed in the paper. However, since only a small fraction of anterior nuclei showed three or four bright FISH spots, it is unclear whether *hb* replication occurred in all nuclei at the time of observation (fixation). For single-locus analysis, we followed a default criterion to roughly determine

the gene replication status of each nucleus in the embryo. I.e., nuclei with active sister loci pairs are post-replication and have four copies of the *hb* gene, while nuclei without active sister loci pairs were coarsely assumed to be pre-replication and have two copies of the *hb* gene. The number of silent loci in each nucleus was estimated accordingly. In the revised manuscript, the above treatment is described in section “**mRNA quantification**” of the **Methods** part (page 29). Note that our treatment is different from some previous studies (Little et al., 2013; Zoller et al., 2018), which assumed that all nuclei are post-replication. Since only a small fraction of nuclei in our images contained ≥ 3 FISH spots (**Supplementary Fig. 3a**), the expression levels estimated from the two treatments roughly differ by a factor of two. However, such difference should not significantly affect our conclusions about the mean promoter behavior in the first four sections of the **Results** part. Nevertheless, to avoid the possible inaccuracy in gene copy inference, we have summed over the nascent signal in each nucleus as an alternative measure of promoter activity in the revised manuscript. With this single-nucleus data set, we have redone all the mean-level analysis in the first four sections of the **Results** part. Other than a doubling of the expression level, the new results look similar to previous ones. In the fourth section of the **Results** part, the new results support our conclusions better (see our response to Reviewer #1’s comment #6).

Unlike the mean-level analysis, miscounting the number of gene copies can potentially affect the statistical analysis of single-locus data. For example, suppose a nucleus with < 3 FISH spots is indeed post-replication. Then, an observed FISH spot may correspond to a pair of closely located sister loci that are indistinguishable under the microscope (Little et al., 2013; Zoller et al., 2018). In such a case, the distribution of the observed FISH signal, $P(m_{\text{ob}})$, should be the convolution of that of a single gene locus, $P(m_{\text{single}})$, i.e.,

$$P(m_{\text{ob}}) = P(m_{\text{single}}) * P(m_{\text{single}}) \quad (1)$$

When fitting the intensity distribution of FISH spots, we assumed that a certain percentage (α) of the observed FISH spots are in such a case. The total distribution was written as

$$P(m_{\text{ob}}) = (1 - \alpha)P(m_{\text{single}}) + \alpha P(m_{\text{single}}) * P(m_{\text{single}}) \quad (2)$$

By fitting the experimental data, we estimated that $\sim 50\%$ of the observed FISH spots were composed of indistinguishable sister loci pairs. In the original manuscript, the above content was described in section “**Modeling the DNA replication effect**” of **Supplementary Note**. In the revised manuscript, we have included more details in the description (now in **Supplementary Note 6.9** and **6.10**).

Regarding the distance threshold used to identify sister loci pairs, we chose it based on the shape of the distribution of locus-locus distance. Specifically, by computing the mutual distance between every possible pair of bright FISH spots in each nucleus, we found that the distribution of these mutual distances exhibited two distinct populations (**Supplementary Fig. 2d**). The population with smaller mutual distances corresponds to sister loci pairs, while the other population corresponds to

unpaired loci. Thus, we chose a distance threshold of 0.71 μm that lay at the valley point between the two populations to identify sister loci pairs. In the revised manuscript, we have added the above description of threshold choice in section “**mRNA quantification**” of the **Methods** part (pages 28–29).

We apologize for not mentioning the complexity of early replication and the potential presence of three or four spots in the original version of the main text. In the revised manuscript, we have added a description in the second paragraph of the **Results** part (page 6), i.e.:

For the CDS signal, we observed bright FISH spots in the anterior part of nc11–13 embryos (Fig. 1c). Most anterior nuclei contained two bright spots, while some nuclei exhibited three or four spots due to the replication of the hb gene (Supplementary Fig. 3a).

4. To compare signals from different probes, it is essential to verify that the efficiency of fluorescent detectors are comparable. It might be useful to switch fluorescent detectors between P1 and P2.

Response:

Following the reviewer’s comment, we have done a control experiment using P1 and P2 probes with switched fluorophores (TAMRA for P2-3’UTR probes and Alexa Fluor™ 647 for P1-5’UTR probes). Neither the percentage of active nuclei nor the maximal transcription level of the anterior expression domain shows a significant difference from the previous results (**Supplementary Fig. 2e, f**). Thus, the efficiencies of fluorescent detectors are comparable. In the revised manuscript, the above results are briefly presented in section “**smFISH probe design**” of the **Methods** part (pages 24–25).

Modeling :

5. The choice for a 3-state model is not very clearly explained (line 251)

In addition, why a 3-state model with 2 ON/1 OFF is envisaged rather than 2 OFF and 1 ON state (as shown in the context of Drosophila embryos in Pimmitt, Dejean et al., 2021)?

Indeed, while it’s easy to envisage various biochemical state for inactive states, what would distinguish the two proposed active states?

This point could be discussed

Response:

We apologize for not explaining the model selection clearly in the original manuscript. The choice for a three-state model was based on the shape of the distributions of nascent mRNA signals. The basic logic may be explained as follows:

For a general N -state model of transcription kinetics, the probability distribution of nascent mRNA signal m satisfies a master equation

$$\frac{d\mathbf{P}(m)}{d\tau} = (\mathbf{K} - \mathbf{K}_{\text{INI}})\mathbf{P}(m) + \mathbf{K}_{\text{INI}}\mathbf{P}(m - g(\tau)) \quad (3)$$

where $\mathbf{P}(m) = \begin{bmatrix} P(0, m) \\ P(1, m) \\ \vdots \\ P(N-1, m) \end{bmatrix}$ denotes the nascent mRNA distributions of each transcription state,

$\mathbf{K} = \begin{bmatrix} -\sum_{i \neq 0} k_{0,i} & k_{10} & \cdots & k_{N-1,0} \\ k_{01} & -\sum_{i \neq 1} k_{1,i} & \cdots & k_{N-1,1} \\ \vdots & \vdots & \ddots & \vdots \\ k_{0,N-1} & k_{1,N-1} & \cdots & -\sum_{i \neq N-1} k_{i,N-1} \end{bmatrix}$ describes transitions between different promoter states, and

$\mathbf{K}_{\text{INI}} = \begin{bmatrix} k_{\text{INI},0} & 0 & \cdots & 0 \\ 0 & k_{\text{INI},1} & \cdots & 0 \\ \vdots & \vdots & \ddots & \vdots \\ 0 & 0 & \cdots & k_{\text{INI},N-1} \end{bmatrix}$ represents transcription initiation of each promoter state (Xu et al.,

2015; Xu et al., 2016). In an extreme case of slow state-transitions ($\mathbf{K}T_{\text{RES}} \approx 0$, with T_{RES} the nascent mRNA residence time), each row in Equation (3) is decoupled into a 1-state equation. The steady-state solution of each 1-state equation is a unimodal distribution peaked at $m_{\text{peak},i} \propto k_{\text{INI},i}$ (Xu et al., 2015; Xu et al., 2016; Zenklusen et al., 2008). Thus, the steady-state solution of Equation (3) is a linear combination of these unimodal distributions weighted by the probability of each transcription state. Obviously, for a system with N distinct $k_{\text{INI},i}$, the solution is a multimodal distribution with N peaks, each of which corresponds to a transcription state. If $k_{\text{INI},i}$ of some states are identical, the number of peaks reduces accordingly. Moreover, outside the slow-transition region of the parameter space, coupling between rows in Equation (3) tends to merge peaks and further reduces the distribution modality (Innocentini Gda et al., 2013; Xu et al., 2016). Consequently, the steady-state solution of an N -state model with M ($M \leq N$) distinct transcription initiation rates has M peaks at most.

In this paper, we applied the above result as a criterion for model selection. Specifically, we observed that both P1 and P2 nascent mRNA signals from individual promoter loci exhibited a trimodal distribution with a peak at $m = 0$ and two distinct peaks at $m > 0$ (Fig. 5a, b). The simplest kinetic model to explain this result is a three-state model composed of an inactive state ($k_{\text{INI},0} = 0$) and two active states with distinct transcription initiation rates ($k_{\text{INI},1} \neq k_{\text{INI},2} > 0$). As the reviewer pointed out, this model is different from the three-state models proposed in some other papers (e.g., (Bintu et al., 2016; Pimmitt et al., 2021)), which contained two inactive states and one active state. Those “2 OFF/1 ON” models can describe fine steps within the inactive phase of gene regulation. However, their nascent mRNA distributions were either unimodal or bimodal. I.e., they cannot

reproduce the trimodal distribution observed in this paper. In fact, when fitting the experimental distributions, we used a more general model, in which $k_{INI,1}$ and $k_{INI,2}$ were allowed to reach zero (See Supplementary Note 6.10). The inferred $k_{INI,1}$ and $k_{INI,2}$ were always nonzero, indicating that our “2 ON/1 OFF” model is the right choice for *hb* promoters.

By applying the above analysis to enhancer deletion experiments, we further showed that, for both P1 and P2, the two active promoter states correspond to different Bcd binding configurations of the two enhancers. The weaker active state (state 1) is driven by Bcd binding at a single enhancer, while the stronger active state (state 2) is driven by Bcd binding at both enhancers. In contrast, genes studied in many previous papers have simple *cis*-regulatory sequences (i.e., one enhancer), for which a model with one active state may be good enough. It should be noted that both the “2 ON/1 OFF” and “2 OFF/1 ON” models are simplifications of the actual transcription process, which involves far more molecular steps. Since our paper focuses on the many-body interactions between enhancers and promoters, whose primary feature has been captured by multiple active states, it may be valid to neglect detailed kinetic steps within the OFF state.

In the original manuscript, we briefly described the rationale of model selection in section “**Model selection**” of Supplementary Note and discussed the biological meaning of each active state in the last two paragraphs of the **Results** part. To be more comprehensive, we have improved the statements in **Results** (the second paragraph of the last section, page 16) and Supplementary Note (sections **6.1** and **6.3**) to include all content of this answer.

6. Regarding promoter regulation by more than 2 states, other references could be considered Innocentini et al., 2013 ; Tantale et al., 2016; Pimmett et al., 2021.

Response:

We thank the reviewer for providing these references. In the revised manuscript, we cite them in the fifth paragraph of the **Discussion** part (page 21), and Supplementary Note 6.1 and 6.3.

Minor points:

7. The last sentence of the abstract line is more strongly phrased than the conclusions from the data allow. Since the entire manuscript solely uses fixed imaging that doesn't capture the dynamic aspects of gene regulation, line 27 should be reformulated

Response:

We agree that fixed imaging does not directly provide temporal information of gene regulation. However, the stochastic dynamics of individual gene loci leads to fluctuations in an ensemble of single-locus data, whose statistical properties may be used to extract certain aspects of the

regulation dynamics. In this paper, by analyzing the distribution of nascent mRNA signals, we found a general scheme of regulation dynamics for the two *hb* promoters. Based on this scheme, we quantitatively inferred key kinetic rates of the regulation process. These aspects provide essential information of the underlining mechanisms of gene regulation. Yet, they may not be confused with the actual dynamics of gene regulation, which can only be captured from live samples. To avoid ambiguity, we have reformulated the last sentence of the abstract as follows:

These results provide a quantitative framework for understanding the kinetic mechanisms of complex eukaryotic gene regulation.

8. Line 125 : mention in the main text that the Fano factor is calculated for nascent mRNAs and not on cytoplasmic mRNA.

Response:

We regret this ambiguity. The sentence is now restated as follows:

The Fano factors of anterior P1- and P2-specific nascent signals were much larger than one.

9. Sup1 b : there is a '19' number in the middle of the graph

Response:

We apologize for our negligence. In the revised manuscript, this number has been removed from the figure panel (now renumbered **Supplementary Fig. 3c**)

10. Name with a panel the FISH image + quantification in the bottom part of Figure 1b. Explain more in the figure legend how the data were 'binned'.

Response:

Following the reviewer's comment, we have added a title for each panel of **Fig. 1b** and explained how the data were binned in more detail in the figure legend as follows:

The single-locus data were binned along the AP axis (bin size: 0.05 EL, step size: 0.025 EL).

We have added similar explanations to the legends of all figures with binning.

11. Can the authors explain why the residency time for P1 transcript is 3 times longer than that of P2 transcript. The supp method section (page 4) indicates $T_r = 142$ for P1 and $T_r = 46$ sec for P2. Where do these estimates come from?

Response:

In this paper, we estimated the post-elongation residence time (T_R) for P1- and P2-driven transcripts from the ratios between the CDS and promoter-specific signals. According to our model of transcriptional kinetics, each nascent mRNA molecule stays on the gene locus for a fixed period $T_{RES} = L/V_{EL} + T_R$, where L/V_{EL} is the elongation time and T_R is the post-elongation residence time. The FISH signal of a promoter locus is the sum of signals from all nascent mRNAs initiated within T_{RES} before the observation time, i.e., $m = \sum_{-T_{RES} \leq \tau \leq 0} g(\tau)$. Here the signal from a single nascent mRNA initiated at time τ is described by a contribution function $g(\tau)$, whose shape depends on the target positions of the probe set and the magnitude of T_R . Specifically, $g(\tau) = 1$ for $-T_{RES} \leq \tau \leq -L/V_{EL}$, while it decreases from one to zero for $-L/V_{EL} < \tau \leq 0$ (**Supplementary Fig. 5b**).

For two probe sets targeting nascent mRNAs of the same promoter, their mean signals are proportional to the time average of their contribution functions, i.e.,

$$\frac{\langle m_1 \rangle}{\langle m_2 \rangle} = \frac{\bar{g}_1}{\bar{g}_2} \quad (4)$$

where $\bar{g} = \frac{1}{T_{RES}} \int_{-T_{RES}}^0 g(\tau) d\tau$ denotes time averaging, and g_1 and g_2 are contribution functions of the two probe sets, respectively. We notice that the time average of each contribution function can be divided into two parts, i.e.,

$$\bar{g} = \frac{1}{T_{RES}} \left(T_R + \int_{-L/V_{EL}}^0 g(\tau) d\tau \right) = \frac{1}{T_{RES}} \left(T_R + \frac{L}{V_{EL}} \bar{g}_0 \right) \quad (5)$$

The second term in parentheses may be defined as the time average of the probes' null contribution function (g_0) with $T_R = 0$. Since g_0 is independent of T_R , Equation (5) can be rewritten as

$$\frac{\langle m_1 \rangle}{\langle m_2 \rangle} = \frac{\bar{g}_{10} \cdot L + T_R V_{EL}}{\bar{g}_{20} \cdot L + T_R V_{EL}} \quad (6)$$

Or

$$T_R = \frac{L(\bar{g}_{10} - a\bar{g}_{20})}{V_{EL}(a-1)} \quad (7)$$

where $a = \frac{\langle m_1 \rangle}{\langle m_2 \rangle}$ is experimentally measurable (see the second section of the **Results** part, page 9).

Equation (7) may be used to estimate T_R for P1- and P2-driven transcripts. For P1-driven transcripts, the null contribution functions of the CDS and P1-5'UTR probes satisfy $\overline{g_{P1-CDS,0}} = 0.2613$ and $\overline{g_{P1-5'UTR,0}} = 0.9613$, while the ratio between their nascent signals was measured to be $a_1 = 0.53$. Thus, with $L = 6334$ bp and $V_{EL} = 1.5$ kb/min, we estimated $T_{R-P1} = 142$ s. For P2-driven transcripts, the null contribution functions of the CDS and P2-3'UTR probes satisfy $\overline{g_{P2-CDS,0}} = 0.5977$ and $\overline{g_{P2-3'UTR,0}} = 0.0514$. Different from a_1 , we have re-measured the ratio between P2-CDS and P2-3'UTR nascent signals in the revised manuscript following Reviewer #1's comment. The new result is $a_2 = 3.72$, larger than the previous value. With $L = 3621$ bp and $V_{EL} = 1.5$ kb/min, we now estimate $T_{R-P2} = 22$ s. This value is smaller than the last version ($T_{R-P2} = 46$ s), and it agrees better with the previous study (<35 s (Zoller et al., 2018)).

In the original manuscript, we described the above content in section "**Mean, variance, and noise**" of **Supplementary Note**. This section also included some other contents. To be more specific and avoid confusion, we have divided this section into four parts in the revised manuscript. Section "**6.5 Estimating the post-elongation residence time**" now describes the estimation of T_R .

So far, it is unclear why T_{R-P1} is much longer than T_{R-P2} . One possibility is that the termination of P1-driven transcripts may be interrupted by the dominant P2-driven transcripts. As a test of this hypothesis, we have done a control experiment using CRISPR mutant lines with P1 and P2 deletions ($\Delta P1C$ and $\Delta P2C$ (Ling et al., 2019), obtained from Dr. Stephen Small (New York University) and Dr. Pinar Onal (Northwestern University)). By labeling these mutant embryos with P1-5'UTR, P2-3'UTR, and CDS probes, we confirm that only one promoter is functional in each mutant line (**Supplementary Fig. 4a**). For either mutant, the average signal of transcript-specific probes is proportional to that of CDS probes in the anterior expression domain (**Supplementary Fig. 4b**). Specifically, we estimate $a_2 = 5.56 \pm 0.86$ for $\Delta P1C$ and $a_1 = 0.39 \pm 0.02$ for $\Delta P2C$, which lead to $T_{R-P2} = 10$ s for $\Delta P1C$ and $T_{R-P1} = 47$ s for $\Delta P2C$. Both T_{R-P1} and T_{R-P2} in single-promoter embryos are shorter than in the wild-type embryo. This phenomenon supports the idea that transcripts from the two promoters may interrupt each other's termination process. The remaining difference between T_{R-P1} and T_{R-P2} in single-promoter embryos may be intrinsic since P1- and P2-driven transcripts have different termination sites. In the revised manuscript, this control experiment is presented in **Supplementary Note 3**.

References

- Bintu, L., Yong, J., Antebi, Y.E., McCue, K., Kazuki, Y., Uno, N., Oshimura, M., and Elowitz, M.B. Dynamics of epigenetic regulation at the single-cell level. *Science* **351**, 720-724 (2016).
- Bothma, J.P., Garcia, H.G., Ng, S., Perry, M.W., Gregor, T., and Levine, M. Enhancer additivity and non-additivity are determined by enhancer strength in the *Drosophila* embryo. *Elife* **4**, e07956 (2015).
- Cheung, D., Miles, C., Kreitman, M., and Ma, J. Scaling of the Bicoid morphogen gradient by a volume-dependent production rate. *Development* **138**, 2741-2749 (2011).
- Driever, W., and Nusslein-Volhard, C. The bicoid protein is a positive regulator of *hunchback* transcription in the early *Drosophila* embryo. *Nature* **337**, 138-143 (1989).
- Garcia, H.G., Tikhonov, M., Lin, A., and Gregor, T. Quantitative imaging of transcription in living *Drosophila* embryos links polymerase activity to patterning. *Curr. Biol.* **23**, 2140-2145 (2013).
- Gregor, T., Wieschaus, E.F., McGregor, A.P., Bialek, W., and Tank, D.W. Stability and nuclear dynamics of the bicoid morphogen gradient. *Cell* **130**, 141-152 (2007).
- Houchmandzadeh, B., Wieschaus, E., and Leibler, S. Establishment of developmental precision and proportions in the early *Drosophila* embryo. *Nature* **415**, 798-802 (2002).
- Hulskamp, M., Pfeifle, C., and Tautz, D. A morphogenetic gradient of *hunchback* protein organizes the expression of the gap genes *Kruppel* and *knirps* in the early *Drosophila* embryo. *Nature* **346**, 577-580 (1990).
- Innocentini Gda, C., Forger, M., Ramos, A.F., Radulescu, O., and Hornos, J.E. Multimodality and flexibility of stochastic gene expression. *Bull. Math. Biol.* **75**, 2600-2630 (2013).
- Lim, B., Fukaya, T., Heist, T., and Levine, M. Temporal dynamics of pair-rule stripes in living *Drosophila* embryos. *Proc. Natl. Acad. Sci. U S A* **115**, 8376-8381 (2018).
- Ling, J., Umezawa, K.Y., Scott, T., and Small, S. Bicoid-Dependent Activation of the Target Gene *hunchback* Requires a Two-Motif Sequence Code in a Specific Basal Promoter. *Mol. Cell* **75**, 1178-1187 (2019).
- Little, S.C., Tikhonov, M., and Gregor, T. Precise developmental gene expression arises from globally stochastic transcriptional activity. *Cell* **154**, 789-800 (2013).
- Liu, F., Morrison, A.H., and Gregor, T. Dynamic interpretation of maternal inputs by the *Drosophila* segmentation gene network. *Proc. Natl. Acad. Sci. U S A* **110**, 6724-6729 (2013).
- Liu, J., and Ma, J. Modulation of temporal dynamics of gene transcription by activator potency in the *Drosophila* embryo. *Development* **142**, 3781-3790 (2015).
- Lucas, T., Ferraro, T., Roelens, B., De Las Heras Chanes, J., Walczak, A.M., Coppey, M., and Dostatni, N. Live imaging of bicoid-dependent transcription in *Drosophila* embryos. *Curr. Biol.* **23**, 2135-2139 (2013).
- Margolis, J.S., Borowsky, M.L., Steingrimsson, E., Shim, C.W., Lengyel, J.A., and Posakony, J.W. Posterior stripe expression of *hunchback* is driven from two promoters by a common enhancer element. *Development* **121**, 3067-3077 (1995).
- Mir, M., Reimer, A., Haines, J.E., Li, X.Y., Stadler, M., Garcia, H., Eisen, M.B., and Darzacq, X. Dense Bicoid hubs accentuate binding along the morphogen gradient. *Genes Dev.* **31**, 1784-1794 (2017).

Mir, M., Stadler, M.R., Ortiz, S.A., Hannon, C.E., Harrison, M.M., Darzacq, X., and Eisen, M.B. Dynamic multifactor hubs interact transiently with sites of active transcription in *Drosophila* embryos. *Elife* **7**, e40497 (2018).

Park, J., Estrada, J., Johnson, G., Vincent, B.J., Ricci-Tam, C., Bragdon, M.D., Shulgina, Y., Cha, A., Wunderlich, Z., Gunawardena, J., *et al.* Dissecting the sharp response of a canonical developmental enhancer reveals multiple sources of cooperativity. *Elife* **8**, e41266 (2019).

Perry, M.W., Boettiger, A.N., and Levine, M. Multiple enhancers ensure precision of gap gene-expression patterns in the *Drosophila* embryo. *Proc. Natl. Acad. Sci. U S A* **108**, 13570-13575 (2011).

Pimmett, V.L., Dejean, M., Fernandez, C., Trullo, A., Bertrand, E., Radulescu, O., and Lagha, M. Quantitative imaging of transcription in living *Drosophila* embryos reveals the impact of core promoter motifs on promoter state dynamics. *Nat. Commun.* **12**, 4504 (2021).

Struhl, G., Johnston, P., and Lawrence, P.A. Control of *Drosophila* body pattern by the *hunchback* morphogen gradient. *Cell* **69**, 237-249 (1992).

Tautz, D., Lehmann, R., Schnurch, H., Schuh, R., Seifert, E., Kienlin, A., Jones, K., and Jackle, H. Finger protein of novel structure encoded by *hunchback*, a second member of the gap class of *Drosophila* segmentation genes. *Nature* **327**, 383-389 (1987).

Wu, X., Vasisht, V., Kosman, D., Reinitz, J., and Small, S. Thoracic patterning by the *Drosophila* gap gene *hunchback*. *Dev. Biol.* **237**, 79-92 (2001).

Xu, H., Sepúlveda, L.A., Figard, L., Sokac, A.M., and Golding, I. Combining protein and mRNA quantification to decipher transcriptional regulation. *Nat. Methods* **12**, 739-742 (2015).

Xu, H., Skinner, S.O., Sokac, A.M., and Golding, I. Stochastic Kinetics of Nascent RNA. *Phys. Rev. Lett.* **117**, 128101 (2016).

Yang, Z., Zhu, H., Kong, K., Wu, X., Chen, J., Li, P., Jiang, J., Zhao, J., Cui, B., and Liu, F. The dynamic transmission of positional information in *stau(-)* mutants during *Drosophila* embryogenesis. *Elife* **9**, e54276 (2020).

Zenklusen, D., Larson, D.R., and Singer, R.H. Single-RNA counting reveals alternative modes of gene expression in yeast. *Nat. Struct. Mol. Biol.* **15**, 1263-1271 (2008).

Zhu, H., Cui, Y., Luo, C., and Liu, F. Quantifying Temperature Compensation of Bicoid Gradients with a Fast T-Tunable Microfluidic Device. *Biophys. J.* **119**, 1193-1203 (2020).

Zoller, B., Little, S.C., and Gregor, T. Diverse Spatial Expression Patterns Emerge from Unified Kinetics of Transcriptional Bursting. *Cell* **175**, 835-847 (2018).

Reviewers' Comments:

Reviewer #1:

Remarks to the Author:

The authors addressed all my concerns on the manuscript. I support the publication of this revised manuscript in Nature Communications.

Reviewer #2:

Remarks to the Author:

The manuscript is significantly improved and overall I am happy that the authors have addressed my all of my concerns. congratulations!